# WHAT'S IN THE BOX? EXPLORING THE INNER LIFE OF NEURAL NETWORKS WITH ROBUST RULES

## ABSTRACT

We propose a novel method for exploring how neurons within a neural network interact. In particular, we consider activation values of a network for given data, and propose to mine noise-robust rules of the form $X \to Y$, where $X$ and $Y$ are sets of neurons in different layers. To ensure we obtain a small and non-redundant set of high quality rules, we formalize the problem in terms of the Minimum Description Length principle, by which we identify the best set of rules as the one that best compresses the activation data. To discover good rule sets, we propose the unsupervised EXPLAINN algorithm. Extensive evaluation shows that our rules give clear insight in how networks perceive the world: they identify shared, resp. class-specific traits, compositionality within the network, as well as locality in convolutional layers. Our rules are easily interpretable, but also super-charge prototyping as they identify which groups of neurons to consider in unison.

## 1 INTRODUCTION

Neural networks provide state of the art performance in many settings. How they perform their tasks, how they perceive the world, and especially, how the neurons within the network operate in concert, however, remains largely elusive. While there exists a plethora of methods for explaining neural networks, most focus on the mapping between input and output (e.g. model distillation) or only characterize individual neurons within the network (e.g. prototyping). In this paper, we introduce a new method that explains how the neurons in a neural network interact. In particular, we consider the activations of neurons in the network over a given dataset, and propose to characterize these in terms of rules $X \to Y$, where $X$ and $Y$ are sets of neurons in different layers of the network. A rule hence represents that neurons $Y$ are typically active when neurons $X$ are. For robustness we explicitly allow for noise, and to ensure that we discover succinct and non-redundant rules we formalize the problem in terms of the Minimum Description Length principle. To discover good rule sets, we propose the unsupervised EXPLAINN method. The rules that we discover give clear insight in how the network performs its task. As we will see, they identify what the network deems similar and different between classes, show how information flows within the network, and show which convolutional filters it expects to be active where. Our rules are easily interpretable, give insight in the differences between datasets, show the effects of fine-tuning, as well as super-charge prototyping as they tell which neurons to consider in unison.

Explaining neural nets is of widespread interest, and especially important with the emergence of applications in healthcare and autonomous driving. In the interest of space we here shortly introduce the work most relevant to ours, while for more information we refer to surveys (Adadi & Berrada, 2018; Ras et al., 2018; Xie et al., 2020; Gilpin et al., 2018). There exist several proposals for investigating how networks arrive at a decision for a given sample, with saliency mapping techniques for CNNs among the most prominent (Bach et al., 2015; Zhou et al., 2016; Sundararajan et al., 2017; Shrikumar et al., 2017). Although these provide insight on what parts of the image are used, they are inherently limited to single samples, and do not reveal structure across multiple samples or classes. For explaining the inner working of a CNN, research mostly focuses on feature visualization techniques (Olah et al., 2017) that produce visual representations of the information captured by neurons (Mordvintsev et al., 2015; Gatys et al., 2015). Although these visualizations provide insight on how CNNs perceive the world (M. Øygard, 2016; Olah et al., 2018) it has been shown that concepts are often encoded over multiple neurons, and that inspecting individual neurons does not

provide meaningful information about their role (Szegedy et al., 2013; Bau et al., 2017). How to find such groups of neurons and how the information is routed between layers remains unsolved.

An orthogonal approach is that of model distillation, where we train easy to interpret white box models that mimic the decisions of a neural network (Ribeiro et al., 2016; Frosst & Hinton, 2017; Bastani et al., 2017; Tan et al., 2018). Rules of the form *(if–then)* are easily interpretable, and hence a popular technique for model distillation (Taha & Ghosh, 1999; Lakkaraju et al., 2017). Although some consider individual neurons to determine what rules to return (Robnik-Šikonja & Kononenko, 2008; Özbakır et al., 2010; Barakat & Diederich, 2005) all these techniques only yield rules that directly map input to output, and hence do not provide insight into how information flows through the network. Tran & dAvila Garcez (2018) mine association rules from Deep Belief Networks. Their approach, however, suffers from the pattern explosion in typical in frequency based rule mining, and is not applicable to state of the art networks. Chu et al. (2018) aim to explain piecewise linear NNs utilizing polytope theory to derive decision features of the network. Providing strong guarantees, this approach is however limited to PLNNs of extremely small size ($< 20$ neurons in hidden layers).

We instead propose to mine sets of rules to discover groups of neurons that act together across different layers in feed forward networks, and so reveal how information is composed and routed through the network to arrive at the output. To discover rules over neuron activations, we need an unsupervised approach. While a plethora of rule mining methods exists, either based on frequency (Agrawal & Srikant, 1994; Bayardo, 1998; Moerchen et al., 2011) or statistical testing (Hämäläinen, 2012; Webb, 2010), these typically return millions of rules even for small datasets, thus undermining the goal of interpretability. We therefore take a pattern set mining approach similar to GRAB (Fischer & Vreeken, 2019), where we are after that set of rules that maximizes a global criterion, rather than treating each rule independently. Providing succinct and accurate sets of rules, GRAB is however limited to conjunctive expressions. This traditional pattern language is too restrictive for our setting, as we are also after rules that explain shared patterns between clases, and are robust to the inherently noisy activation data, which both require a more expressive pattern language of conjunctions, approximate conjunctions, and disjunctions. We hence present EXPLAINN, a non-parametric and unsupervised method that discovers sets of rules that can model these types of rules.

## 2 THEORY

We first informally discuss how to discover association rules between neurons. We then formally introduce the concept of robust rules, and how to find them for arbitrary binary datasets, last, we show how to combine these ideas to reveal how neurons are orchestrated within feedforward networks.

### 2.1 PATTERNS OF NEURON CO-ACTIVATION

Similar to neurons in the brain, when they are active, artificial neurons also send information along their outgoing edges. To understand flow of information through the network, it is hence essential to understand the activation patterns of neurons between layers. Our key idea is to use recent advances in pattern mining to discover a succinct and non-redundant set of rules that together describe the activation patterns found for a given dataset. For two layers $I_i, I_j$, these rules $X \rightarrow Y, X \subset I_i, Y \subset I_j$ express that the set of neurons $Y$ are usually co-activated when neurons $X$ are co-activated. That is, such a rule provides us *local* information about co-activations within, as well as the dependence of neurons between layers. Starting from the output layer, we discover rules between consecutive layers $I_j, I_{j-1}$. Discovering overlapping rules between layers $X \rightarrow Y$ and $Y \rightarrow Z, X \subset I_j, Y \subset I_{j-1}, Z \subset I_{j-2}$, allows us to trace how information flows through the entire network.

Before we can mine rules between two sets of neurons – e.g. layers – $I_i$ and $I_j$ of a network, we have to obtain its binarized activations for a given data set $\mathcal{D} = \{d_k = (s_k, o_k)\}$. In particular, for each sample $s_k$ and neuron set $I_i$, we take the tensor of activations $\phi_i$ and binarize it to $\phi_i^b$. For networks with *ReLU* activations, which binarize naturally at threshold 0, we might lose some information about activation strength that is eventually used by subsequent layers, but it allows us to derive crisp symbolic, and directly interpretable statements on how neurons interact. Furthermore, binarization reflects the natural on/off state of biological neurons, also captured by smooth step functions such as sigmoid or tanh used in artifical neural networks. We gather the binarized activations into a dataset

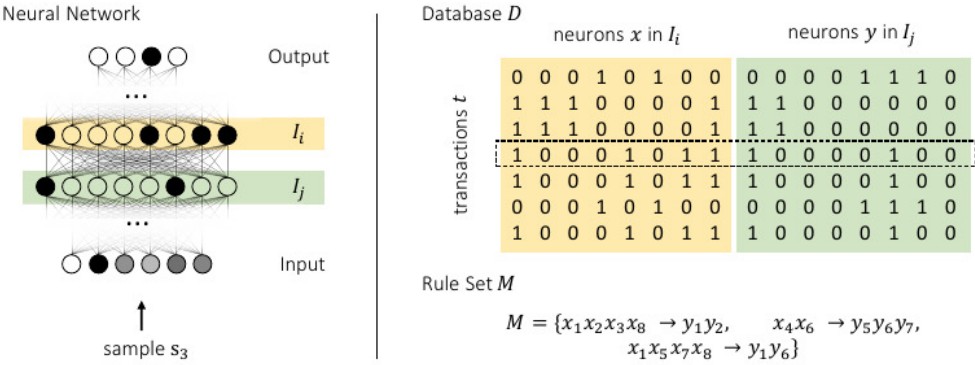

Figure 1: *Overview.* For a given network (left), binarized activations are gathered for the layers $I_i$, $I_j$ for each sample, and summarized in the binary database $D$ (right). Rules are discovered over $D$, where a good rule set $M$ is given with at the bottom right, with rules $X \to Y$, $X \in I_i$, $Y \in I_j$.

$D$ where each row $t_k$ corresponds to the concatenation of $\phi_i^b$ and $\phi_j^b$ of $I_i$ and $I_j$ for $s_k$, i.e., $t_k \in D$ is a binary vector of length $|I_i| + |I_j|$. See Fig. 1 for a toy example.

Next, given $D$, our goal is to find that set of rules that together succinctly describe the observed activations. The Minimum Description Length (MDL) principle lends itself as an objective to find such sets. MDL is a statistically well-founded and computable approximation of Kolmogorov Complexity (Li & Vitányi, 1993). First introduced by Rissanen (1978), the essential idea is that the model $M^* \in \mathcal{M}$ that best describes data $D$ is the model that losslessly encodes $D$ using the fewest bits $M^* = \arg\min_{M \in \mathcal{M}} L(D, M)$. Here, our model class $\mathcal{M}$ is the superset of all possible rules over $D$, and by MDL we identify the best model $M^*$ as the one that compresses the data best. Traditionally, rule mining is restricted to conjunctions over items, which is not sufficient for our application; neuron activations tend to be noisy, labels are inherently mutually exclusive, and hence we consider an extended language that allows for partial disjunctions of items (neurons, labels) and introduce a codelength function $L(D, M)$ to instantiate MDL for our model class of rule sets.

## 2.2 MDL FOR ROBUST RULES

Our goal is to find a set of rules $M$ that, in terms of description length $L(D, M)$, best describes a binary database $D = \{t \mid t \subset \mathcal{I}\}$ that consists of transactions $t$ that are subsets of items $\mathcal{I}$. Each rule is of the form $X \to Y$, $X, Y \subset \mathcal{I}$, and indicates that $Y$ is strongly associated with, i.e. occurs mostly in transactions where $X$ is present. We say a rule $X \to Y$ *applies* to a transaction $t$ iff $X \subset t$ and say a rule *holds* for $t$ if additionally $Y \subset t$. We indicate these transactions sets as $T_X = \{i \mid t_i \in D.\ X \subset t_i\}$, respectively $T_{Y|X} = \{i \mid t_i \in T_X.\ Y \subset t_i\}$. Based on these definitions of rule transaction sets, we can now formally introduce our codelength function $L(D, M)$.

**Baseline model** Our base model $M_{ind} = \{\emptyset \to I \mid \forall I \in \mathcal{I}\}$ consists of singleton rules only, i.e. it models that all items $\mathcal{I}$ are generated independently. To send the $n$ transactions of $D$ using $M_{ind}$, we simply send for each item $I$ in which out of all transactions in the database it appears. We can do so optimally using a log binomial code, which is given by $\log \binom{|T_\emptyset|}{|T_{I|\emptyset}|} = \log \binom{n}{|T_I|}$. To unambiguously decode, the recipient needs to know each $|T_I|$, which we can optimally encode via the parametric complexities of the binomials, which are defined as $L_{pc}(n) = \log \left( \sum_{k=0}^{n} \frac{n!}{(n-k)!k!} \left( \frac{k}{n} \right)^k \left( \frac{n-k}{n} \right)^{n-k} \right)$, and can be computed in linear time (Kontkanen & Myllymäki, 2007). We thus have $L(D, M_{ind}) = \sum_{I \in \mathcal{I}} \left( \log \binom{n}{|T_I|} + L_{pc}(n) \right)$. $M_{ind}$ serves as our baseline model, and its singleton rules are a required part of any more complex model as they ensure we can always send any data over $\mathcal{I}$.

**Non-trivial models** A non-trivial model $M$ contains rules of the form $X \to Y$, $X, Y \subset \mathcal{I}$ that are not part of $M_{ind}$. The idea is that we first transmit the data for where these non-trivial rules hold, and then send the remaining data using $M_{ind}$. To determine where such a rule applies, the receiver

needs to know where $X$ holds, and hence the data over $X$ needs to be transmitted first. To ensure that we can decode the data, we only consider models $M$ for which the directed graph $G = (\mathcal{I}, E)$ is acyclic, in which there exists an edge between two items $i_1, i_2$ iff they occur in the head and tail of a rule, that is $E = \{(i_1, i_2) \mid \exists X \to Y \in M. \, i_1 \in X \wedge i_2 \in Y\}$. We thus get a codelength

$$L(D \mid M \cup M_{ind}) = \left( \sum_{X \to Y \in M} \log \binom{|T_X|}{|T_{Y|X}|} \right) + \sum_{\emptyset \to I \in M_{ind}} \log \binom{n}{|T_I'|},$$

where $T_I' = \{t \in D \mid (I \in t) \wedge (\forall X \to Y \in M. \, I \in Y \implies t \notin T_{Y|X})\}$ is a modified transaction set containing all transactions with item $I$ not covered by any non-trivial rule.

In addition to the parametric complexities of the binomial codes, the model cost of a non-trivial model also includes the cost of transmitting the non-trivial rules. To transmit a rule $X \to Y$, we first send the cardinalities of $X$ resp. $Y$ using the universal code for integers $L_{\mathbb{N}}$ (Rissanen, 1983). For $n \geq 1$, this is defined as $L_{\mathbb{N}}(z) = \log^* z + \log c_0$ with $\log^*(z) = \log z + \log \log z + \dots$, summing only over the positive components (Rissanen, 1983). To satisfy the Kraft equality we set $c_0 = 2.865064$. Knowing the cardinalities, we can then send the items of $X$ resp. $Y$ one by one using an optimal prefix code given by $L(X) = -\sum_{x \in X} \log \frac{|T_x|}{\sum_{I \in \mathcal{I}} |T_I|}$. For a particular rule $X \to Y \in M$, the model costs for a rule, respectively the full model thus amount to

$$L(X \to Y) = L_{\mathbb{N}}(|X|) + L_{\mathbb{N}}(|Y|) + L(X) + L(Y) + L_{pc}(|T_X|) \, ,$$

$$L(M \cup M_{ind}) = |\mathcal{I}| \times L_{pc}(n) + L_{\mathbb{N}}(|M|) + \sum_{X \to Y \in M} L(X \to Y) \, .$$

We provide an example in App. A.1. With these definitions, we have an MDL score that identifies the best rule set $M^*$ for data $D$ as

$$M^* = \underset{M \in \mathcal{M}}{\arg\min} \left( L(M \cup M_{ind}) + L(D \mid M \cup M_{ind}) \right) \, ,$$

where $\mathcal{M}$ is the space of all possible rule sets over the items in $D$.

**Robust Rules**  In real world applications, which are inherently noisy, we need a score that is robust and can approximate where rules hold and apply. In addition, as output neurons are only active mutually exclusively, rules need to be able to model disjunctions. The key problem with noisy data is that a single missing item in a transaction can cause a whole rule not to hold or apply. To discover rules that generalize well, we need to explicitly account for noise. The idea is to let rules apply, and hold, approximately. Specifying how many items $l$, and $k$, out of all items in the rule head, respectively tail, need to be part of a transaction, we relax the original rule definition to account for random noise (see App. Figs. 7, 8). Setting $l = 1$ and $k = 1$ coincidentally corresponds to a disjunction of items in the head and tail of the rule $X \to Y$, thus allowing to model output neurons correctly, and $l = |X|$ and $k = |Y|$ correspond to the original stringent rule definition. Varying between the two extremes accounts for varying levels of noise. The optimal $l$ and $k$ are those that minimize the MDL score. To ensure a lossless encoding, we need to make sure that the receiver can reconstruct the original data. Thus, for the relaxed head definition, for each rule we send the number of items $l$ for the rule to apply using $L_{\mathbb{N}}(l)$ bits. Knowing each $l$, the receiver can reconstruct where each rule applies. Sending where a rule holds now leaves the receiver with an approximation of the data. To be able to reconstruct the actual data, Fischer & Vreeken (2019) introduced error matrices that when XORed with the approximation yield the original data. These two matrices $\mathcal{X}_{X \to Y}^+$, and $\mathcal{X}_{X \to Y}^-$ correct for the errors made in the part where the rule applies and holds, respectively applies but does not hold (for an example see App. Fig. 7). These error matrices are part of the model $M$ and have to be transmitted with an adapted $L(D, M)$, which we review in App. A.2.

**Complexity of the search**  To discover rules over the activations of two layers $I_i, I_j$, we have to explore all rules formed by subsets of neurons in $I_i$ for the head, combined with any subset of neurons of $I_j$ for the tail. There exist $2^{|I_i|} \times 2^{|I_j|}$ such rules, and hence $2^{2^{|I_i|+|I_j|}}$ distinct models would need to be explored. Fischer & Vreeken (2019) showed that the rule set search space does not lend itself to efficient search as it is neither monotone nor submodular, the counterexamples also holding for our model definition. In fact, for robust rules, we additionally have to consider where rules should apply respectively hold – optimizing $k$ and $l$ – which results in approximately $2^{|I_i| \times |I_j| \times 2^{|I_i|+|I_j|}}$ models (detailed derivation in App. A.4). Exhaustive search is therewith infeasible, which is why we present EXPLAINN, a heuristic algorithm to efficiently discover good sets of rules.

## 2.3 DISCOVERING GOOD RULE SETS WITH EXPLAINN

EXPLAINN is based on the idea of iteratively refining the current model by merging and refining already selected rules. The key insight of the algorithm is that for a rule $X \to Y$ to summarize the data well, also rules $X \to Y'$ with only part of the tail, $Y' \subset Y$, should summarize well, as all tail items should be similarly co-occuring with head $X$. Starting from the baseline model $M_{ind}$ we iteratively and greedily search for better models until we can no longer improve the MDL score. As search steps we consider either introducing a new rule to $M$, by taking a good set of items $X \subset I_i$ for the head and a single item $A \in I_j$ for the tail and refine the model to $M' = M \oplus \{X \to A\}$, seeing if it decreases the overall MDL costs (Eq. 2.2). Or, we merge two existing rules $r_1 = X \to Y_1 \in M$ and $r_2 = X \to Y_2 \in M$, to form a new rule $r' = X \to Y_1 \cup Y_2$ and refine the model to $M' = M \oplus \{r'\} = (M \setminus \{r_1, r_2\}) \cup \{r'\}$. For a rule $r'$, the refinement operator $\oplus$ is adding the rule $r' = X \to Y$ to $M$, removing the merged rules that led to $r'$, if any, and updating the singleton transaction lists $T_A$ for all items $A \in Y$, removing all transactions where $r'$ holds.

To make this search scheme applicable to the scale of a typical neural net, we have to discuss how to efficiently search for candidate rules, with heads that can express anything from conjunctions to disjunctions. Immediately after, we present the full algorithm EXPLAINN for mining high quality rule sets for two arbitrary sets of neurons (e.g. layers) of a network.

**Searching for candidates** A key component of EXPLAINN is the candidate generation process, which implements the two possible steps of generating new and merging existing rules. Given two layers $I_i, I_j$, to efficiently discover rules that are both robust to noise, and may include disjunctively active neurons in the head, we can not enumerate all possible rule heads for each individual tail neuron, as this would result in $|I_j| \times 2^{|I_i|}$ many rules. Instead, we keep a list $H_y$ for each item $y \in I_j$, storing all head neurons $x \in I_i$ for which $y$ is frequently active when $x$ is active, that is $\sigma_{x,y} = \frac{|T_x \cap T_y|}{|T_x|} > \theta$, where $\theta$ is a confidence threshold. We consider a rule $X \to Y$ to be good, if when neurons $X$ are active, the neurons $Y$ are also likely to be active, which is directly represented by the confidence $\theta$. With parameter $\mu$ we account for early decisions on rule merging that later hinder us to see a more general trend. The lists are sorted decreasing on $\sigma$. We search in each $H_y$ for the rule with highest gain over all unions of first $t = 1 \ldots |H_y|$ neurons in the list. We add that rule $X \to y$ with highest gain to the candidate list (see App. Alg. 1). To compute the gain, we consider all possible values $k = 1 \ldots |X|$ to determine for which transactions $T_X^k = \{t \in D \mid |X \cap t| \geq k\}$ the rule should robustly apply, where $k = 1$ corresponds to disjunction and $k = |X|$ to conjunction.

For an individual neuron $y$, such a rule would be optimal, but, our goal is to discover groups of neurons that act in concert. To this end we hence iteratively merge rules with *similar* heads – similar, rather than same, as this gives robustness both against noise in the data, as well as earlier merging decisions of the algorithm. For two rules $X_1 \to Y_1$, $X_2 \to Y_2$ with symmetric difference $X_1 \ominus X_2 = (X_1 \setminus X_2) \cup (X_2 \setminus X_1)$, we consider possible candidate rules $X_1 \cup X_2 \to Y_1 \cup Y_2$ and $X_1 \cap X_2 \to Y_1 \cup Y_2$, iff $|X_1 \ominus X_2| \leq \mu$ for some threshold $\mu \in \mathbb{N}$. For example, $\mu = 1$ corresponds to the case that one head has one label more than the other, all other labels are the same.

Both parameters $\theta$ and $\mu$ are simple, yet effective runtime optimizations, the best results with respect to MDL will always be obtained with the largest search space, i.e. with $\theta$ and $\mu$ set to 0, respectively $|X_1| + |X_2|$. Besides impacting run-time, many of those rules may be uninteresting from a user-perspective, $\mu$ and $\theta$ allow to directly instruct EXPLAINN to ignore such rules.

**EXPLAINN** Assembling the above pieces, we arrive at EXPLAINN, which given two sets of neurons $I_i, I_j$ and a database of activations of these neurons, yields a heuristic approximation to the MDL optimal model $M^*$. By first introducing all relevant single neuron rules, it then proceeds by iteratively merging existing rules using the approach described above, until it can achieve no more gain. For efficiency, we separate the generation of the new rules from the merging of existing rules. In practice, this does not harm performance, as we allow merging of similar heads and can thus revert too greedy decisions introduced earlier. Furthermore, by observing that independent rules $X_1 \to Y_1$, $X_2 \to Y_2$, $Y_1 \cup Y_2 = \emptyset$ do not influence each others impact on codelength, we can add all independent rules with the highest respective gain at once. We give pseudocode as App. Alg. 3.

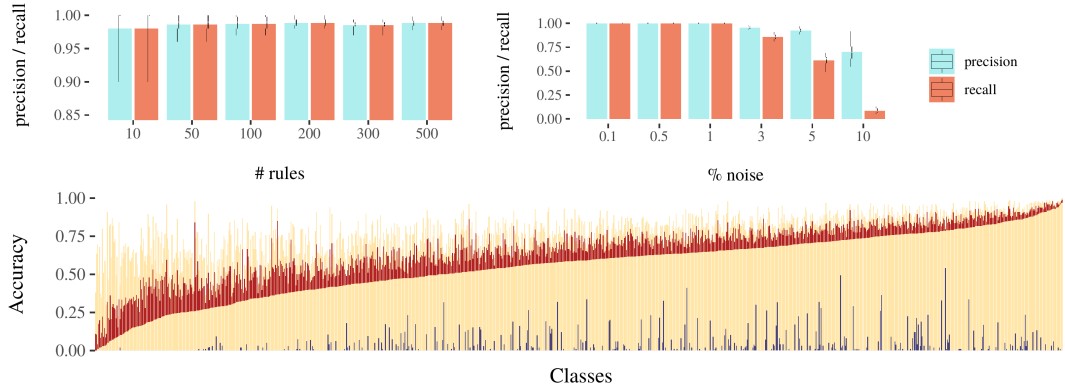

Figure 2: *Evaluation of rule quality.* **Top:** Performance of EXPLAINN as precision and recall on data with varying number of planted rules with mutual exclusive head items (left) and co-occurring head items with varying noise (right). 10% noise corresponds to more noise than signal in the data. We provide the average (bar) and distribution (boxplot) across 10 repetitions. **Bottom:** Accuracy per class of VGG-S before (yellow) and after (blue) intervention on weights connecting neurons to class given by a rule, and 90% quantile of accuracies obtained for randomized intervention (red).

**Complexity of EXPLAINN** The generation of new rules results in time $O(n \times |I_j| \times |I_i|^3)$, by iterating over each neuron in $I_j$, and considering each subset of the most overlapping neurons in $I_i$, and considering each threshold $k = 1 \ldots |I_i|$ for when the rule should apply, and the factor $n$ from intersecting transaction lists $T$ to compute the gain. We can have at most $|I_j|$ generated rules before considering rule merges, and in every iteration of merging we combine two rules, reducing the rule set size by 1. In each such step, we consider $|I_j|^2$ merges, for each of which we compute the gain considering noisy head and tail. We thus have a worst case runtime of $O(n \times |I_j|^4 \times |I_i|)$. As MDL ensures we consider succinct models that capture only the relevant structure in the data, EXPLAINN is in practice much faster and easily scales to several thousands of neurons.

## 3 EXPERIMENTS

In this section we empirically evaluate EXPLAINN on synthetic data with known ground truth and real world data to explore how CNNs perceive the world. We focus on CNNs as they count towards the most widespread use of feedforward networks and naturally lend themselves for visualization, which helps us to interpret the discovered rules. We consider CNNs of different complexity, including GoogLeNet and VGG-S. We compare to traditional prototyping and activation map approaches on *MNIST* (LeCun & Cortes, 2010), and examine which information is used how to arrive at classification for *ImageNet* (Russakovsky et al., 2014). Finally, we investigate the effect of fine-tuning in transfer learning on the Oxford *Flower* data (Nilsback & Zisserman, 2008). We will make the implementation of EXPLAINN publicly available upon publication. For the below experiments, running on commodity hardware EXPLAINN took minutes for *MNIST* and *Flower*, and 6 hours for *ImageNet*, yielding rule sets of hundreds up to 3000 rules, depending on data and choice of layers.

### 3.1 RECOVERING GROUND TRUTH

To evaluate how well EXPLAINN is able to recover the ground truth from data, we first generated synthetic binary data sets of size $n = 10000$ samples initially filled with zeros with $m = \{10, 50, 100, 200, 300, 500\}$ planted rules with up to 5 items in head and tail, respectively. For each rule, the frequency is drawn from $U(.02, .08)$, the confidence is drawn uniformly from $U(.5, 1)$. We introduce noise by flipping 0.1% of the entries chosen uniformly at random. Additionally, 5 noise features with the same frequency as the rule features are introduced. Fischer & Vreeken (2019) showed that such an MDL model works for conjunctive rules, hence we will focus on investigating performance for mutually exclusive rule heads and noise. In the first set of experiments, we set head items to 1 in a mutual exclusive fashion, in line of finding rules over the NN output labels. EXPLAINN achieves high recall and precision (see Figure 2) in terms of retrieving

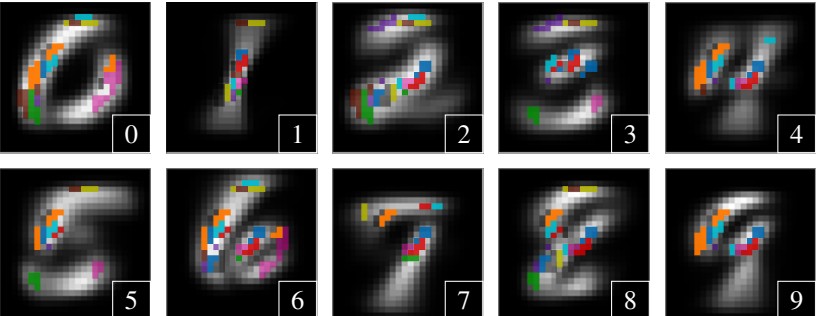

Figure 3: *MNIST* Average activation of neurons for the classes in *MNIST* for filter 2 in the first convolutional layer. Overlayed are the EXPLAINN rules, where pixel groups of the same colour (e.g. purple pixels top left for classes 2, 3) belong to a single rule spanning multiple classes.

exact ground truth rules, and does not retrieve any redundant rules. Next, we investigate the impact of noise on the performance, generating data of $n = 10000$ samples and $m = 100$ rules in a similar setting as above, with head items now initially set co-occuring (conjunctive), but adding varying levels of noise with $\alpha = \{0.1\%, 0.5\%, 1\%, 3\%, 5\%, 10\%\}$ of bitflips in the matrix, where $10\%$ noise means more noise than actual signal. We report the results in Figure 2. As expected by its design, EXPLAINN is fairly robust to noise, even when facing almost the same amount of noise and signal.

### 3.2 TAKING A GLANCE INTO THE REAL WORLD

**How is information filtered**    As a first simple example we consider *MNIST*. We train a simple CNN of 2 convolutional and one fully connected layer using Keras, achieving 99% classification accuracy on test data (see App. B.1 for details). We are interested in what the individual filters learn about the digits, and how EXPLAINN reveals shared features across several classes. We compare to average activation maps and single neuron prototypes. Examining different filters, we observe that average class activation maps do not reveal the purpose of a filter, whereas rules that EXPLAINN discovers identify which pixels together trigger a filter. As an example, consider Fig. 3, in which for filter 2 in layer 1 we visualize the discovered rules on top of the average activation maps (which just represents the original numbers). Whereas the rules identify shared structure (indicated by pixels of the same colour, in the same location, for different classes) such as top-left curve for 0, 4, and 9 (depicted in orange), the prototype (see App. Fig. 9) looks like a maze and does not reveal any insight. Similarly, for filter 36 in layer 2, we directly see from the discovered rules that this filter detects horizontal edges in a class specific manner, whereas prototyping and activation maps again fail to reveal this information (see App. Fig. 10). Interestingly, we also find filters that learn a negative, with activated areas corresponding to background of the digit (see App. Fig. 11).

**How does information flow**    Next, we investigate a more complex setting and examine the activations for the *ImageNet* data set of pretrained VGG-S and GoogLeNet architectures (Chatfield et al., 2014; Szegedy et al., 2015). We here focus on VGG-S, as for this network there exist optimized prototyping methods that yield highly interpretable images for sets of neurons (M. Øygard, 2016). We provide the results for GoogLeNet in App. B.2.1. Starting with rules from the labels to the last layer, we observe that EXPLAINN yields complex rules spanning multiple labels and multiple neurons in the last fully connected layer (FC7), which together encode the information shared between labels. Examples include the faces of certain dog breeds, as well as the coloured beaks of birds (see App. B.2.2). Notably, if we visualize these neurons individually, it is hard to extract anything meaningful from the images: the information is really encoded in the set of neurons that act together (see App. Fig. 14). We also observe cases where rules describe how the network discriminates between very similar classes. We give an example in Fig. 4 for the neurons EXPLAINN discovers to be associated with huskies, with malamutes, and for both classes together. These dog breeds are visually very similar, sharing a black–white fur pattern, as well as head and ear shapes. These traits are reflected by the neurons corresponding to the rule for both breeds. Looking closer, we can see that distinct traits, the more defined snout and ears of the husky, respectively the fluffy fur of the

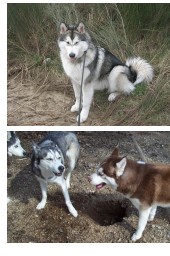

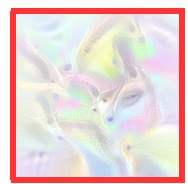

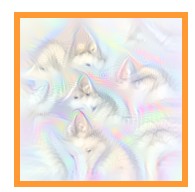

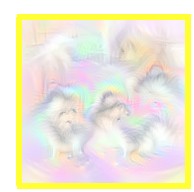

Siberian husky
AND NOT `Malamute`

Siberian husky
AND `Malamute`

`Malamute` AND NOT
Siberian husky

(a) Top: Malamute,
Bottom: Siberian Husky

(b) Rule prototypes for FC7

Figure 4: *Neurons discriminating Huskies and Malamutes.* a) Huskies and Malamutes are very similar looking dog breeds. b) Prototypes for rules $X \to Y$ discovered for classes $X$, `Siberian husky` (red frame), class `Malamute` (yellow frame), resp. both (orange frame) and neurons $Y$ in FC7. The neurons associated with both classes represent typical features shared between the two classes, those associated only with `Siberian huskies` show their slightly sharper, more defined head, while those associated only with `Malamutes` capture their more fluffy fur.

malamute, are picked up by the neurons discovered for the individual dog breeds. Beside discovering what shared and distinct traits the network has learned for classes, we also find out when it learns differences *across* samples of the *same* class. For the dog breed Great Danes, we discover three rules that upon visualization each correspond to visually very different sub-breeds (see App. Fig. 15).

Next we investigate the information flow within the network. Starting with the EXPLAINN rules $X \to Y$ from output layer to last fully connected layer FC7, we apply EXPLAINN to discover rules $Y \to Z$ between FC7 and FC6, where we restrict the heads $Y$ to be groups of neurons found as tails in the previous run. We recursively apply this process until we covered a convolutional layer. This gives us traces of neuronal activity by chaining rules $X \to Y \to Z$ discovered in the iterative runs. We visualize one such trace in Fig. 5, which gives insight in how the network perceives totem poles. We find that the relevant neurons of FC7 and FC6 yield prototypes that show typical totem pole heads with animalistic faces. The neuron sets found for the different filters of the last convolutional layer CONV5, together detect specific parts of the class, including the vertical pole, and the rooftop-like structure, decorated with animalistic shapes with eyes, that is typically found at the top of a totem. We provide another example of an information trace in App. Fig. 16, which shows how the network learns to detect features located typically at certain parts of the image due to the very prevalent side-view photos taken from dogs, a clear sign of overfitting by the neural network.

**Rules carry class information** To determine the quality of a rule set found across classes, we next investigate how the classifier performance changes when we intervene on the neurons in the last fully connected layer that were found associated to a class. For each class $c$, we set incoming weights from neurons $y$ to 0, for which we have discovered a rule $X \to Y, c \in X, y \in Y$, comparing classification rate before and after intervention. To further evaluate the practical impact of these rules, we perform 100 runs intervening on an equally sized random subset of all weights leading to class $c$, again measuring classification rate after intervention. We report the results in Fig. 2 (bottom). For all classes, the performance dropped tremendously, in most cases even to 0. This shows that the discovered rules capture necessary information for classification. Furthermore, we observe that in most cases very related classes are instead predicted, for example `Fire Salamander` to `Spotted Salamander`, `Barbell` to `Dumbbell`, or `Palace` to `Monastery`, which provides interesting opportunities for adversarial attacks.

**The effect of fine tuning** Finally, we show that EXPLAINN can elucidate the effect of fine-tuning in transfer learning. For this we consider Oxford *Flower* data Nilsback & Zisserman (2008), which consists of 8k images of flowers of 102 classes. For investigation, we consider both the vanilla VGG-S network trained on *ImageNet* from above, and a fine-tuned version from the Caffee model zoo[1]. We run EXPLAINN to obtain rules between the output and the final layer of both networks, and give

---

[1]https://github.com/jimgoo/caffe-oxford102

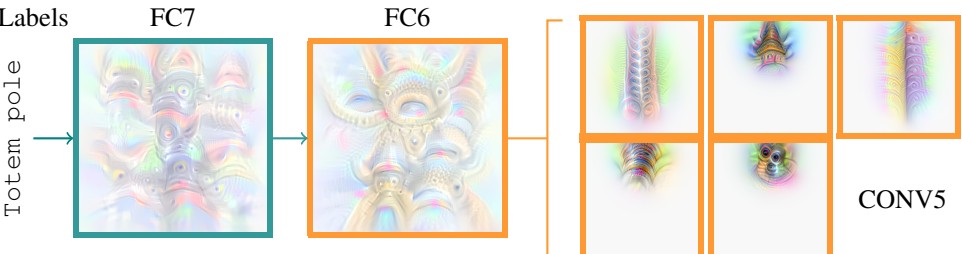

Figure 5: *Information flow for Totem poles.* Example from rule cascades found for *ImageNet* starting at label `Totem pole`. For each rule $X \to Y$, the group of neurons of the tail $Y$ are used to generate a prototype. To discover rule cascades, we start mining rules between output and FC7 layer. Rule tails, which are neurons of FC7, are then used as heads for rules between FC7 and FC6. Similarly, rule tails are used as heads for rules between FC6 and CONV5.

an example in App. Fig. 17. The visualizations show, as expected, a strong emphasize on colour and shape of the corresponding flower class. Interestingly, the visualizations of the same neurons for the original VGG-S show almost identical shapes and pattern, but with less intense colour. Furthermore, we observe some prototypes with animal-like features such as eyes or beaks.

## 4   DISCUSSION AND CONCLUSION

The experiments show that EXPLAINN is uniquely able to discover distinct groups of neurons that *together* capture traits shared and distinct between classes, within-class heterogeneity, and how filters are used to detect shared features, segment background, or detect edges locally. None of these are revealed by activation maps, which miss the local information that patterns provide, nor by saliency maps, which investigate network attention for an individual image. Prototyping is a great tool for visualizing neuron information content, but, by itself is limited by the massive number of possible combinations of neurons, requiring thousands of hours to painstakingly handpick and connect the information of just individual neurons (Olah et al., 2020). Combining EXPLAINN with prototyping permits exploring networks beyond single neurons, by automatically discovering which neurons act in concert, which information they encode, and how information flows through the NN.

In particular, we discover distinct groups of neurons in fully connected layers that capture shared respectively distinct traits across classes, which helps in understanding how the network learns generality but still can discriminate between classes. Due to the local information that our rules provide, we can also detect differences in the perception across samples of a single class, where for example different groups of neurons describe visually different sub-breeds of a class of dogs. By connecting rules that we find across several layers, we can trace how information is gathered and combined to arrive at a classification, from filters that detect typical class specific features in the image, through fully connected layers where multiple neurons together encode the combined information, up to the final classification output. Applying EXPLAINN to investigate the impact of fine-tuning in transfer learning, we found that for the groups of neurons in the given fine-tuned CNN, surprisingly, the contained information is almost identical to the original CNN, but capturing the traits of the new classes almost perfectly. For the given transfer task the fine-tuning thus mostly resulted in routing information differently, rather than having to learn to detect new features.

Overall, EXPLAINN performs well and convincingly answers 'what's in the box?'. It finds surprising results that help to understand how CNNs perceive the world. While many important tasks are solved by such networks, recurrent neural networks play an important role in e.g. language processing. Although rules can likely also help to understand what these models learn, these networks encode an entirely different type of information that is inherently hard to understand and visualize, and hence an exciting challenge for future work. Here, our main interest was characterizing information flow through neural networks, and hence, we focused on subsequent layers. EXPLAINN, however, operates on arbitrary sets of neurons, and hence also naturally allows investigating e.g. residual networks, where the previous *two* layers contribute information. Currently scaling to thousands of neurons, it will make for engaging future work to scale to entire networks at once.

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

## A  MDL FOR ROBUST RULES

In this section we will give extended examples on how to compute the MDL score for a given database and set of rules, elaborate on the error encoding for the rule tails, and give a visual toy example on the impact of the extended pattern language for the rule head.

### A.1  COMPUTING MDL FOR RULES

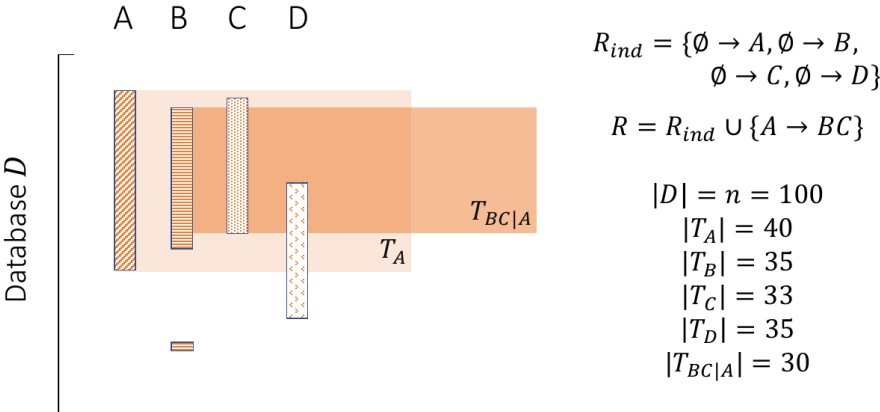

Figure 6: *Example database and model.* A toy database $D$ with blocks indicating where the items $A, B, C, D$ occur in $D$, margins and relevant joint counts are given on the right. A sensible rule set $M \cup M_{ind} = A \to BC \cup M_{ind}$ is given on the right, the part of the database where the rule applies and holds is indicated by a light respectively dark orange area.

For the given example in Fig. 6, we will now compute the codelength $L(D, M) = L(M) + L(D \mid M)$ of transmitting the whole database $D$ using $M \cup M_{ind}$. Here, we will stick with the simple encoding without error matrices, to make the process of computation more understandable. For

reference, we first compute the baseline model, which is given by

$$L(D, M_{ind}) = |\mathcal{I}| \times L_{pc}(|D|) + \sum_{I \in \mathcal{I}} \log \binom{|D|}{|T_I|}$$

$$= 4 \times L_{pc}(100) + \log \binom{100}{40} + 2 \log \binom{100}{35} + \log \binom{100}{33}$$

$$\approx 14.88 + 93.47 + 179.64 + 87.93 = \mathbf{375.92}.$$

Thus, sending the data with just the baseline model costs 375.92 bits. Now, we will compute $L(D, M \cup M_{ind})$, we will start with the costs of sending the data $L(D \mid M \cup M_{ind})$

$$L(D \mid M \cup M_{ind}) = \left( \sum_{X \to Y \in M} \log \binom{|T_X|}{|T_{Y|X}|} \right) + \left( \sum_{I \in \mathcal{I}} \log \binom{|D|}{|T_I'|} \right)$$

$$= \log \binom{40}{30} + \log \binom{100}{40} + \log \binom{100}{5} + \log \binom{100}{3} + \log \binom{100}{35}$$

$$\approx 29.66 + 93.47 + 26.17 + 17.30 + 89.82 = \mathbf{256.42}.$$

The model costs are composed of the parametric complexities for the (adapted) baseline rules, plus the costs of transmitting what the rule is composed of along with its parametric complexity. We thus get

$$L(M \cup M_{ind}) = |\mathcal{I}| \times L_{pc}(|D|) + \sum_{X \to Y \in M} L_{\mathbb{N}}(|X|) + L_{\mathbb{N}}(|Y|) + L(X) + L(Y) + L_{pc}(T_X)$$

$$= 4 \times L_{pc}(100) + L_{\mathbb{N}}(1) + L_{\mathbb{N}}(2) - \log \frac{40}{143} - \log \frac{35}{143} - \log \frac{33}{143} + L_{pc}(40)$$

$$\approx 14.88 + 1.52 + 2.52 + 1.84 + 2.03 + 2.12 + 3.11 = \mathbf{28.02}.$$

Hence, the model with the complex rule has a smaller codelengh than the baseline, with $L(D, M \cup M_{ind}) = \mathbf{284.44}$ bits.

## A.2 THE ERROR ENCODING FOR TAILS

For the error encoding for tails, which allow to discover rules in noisy settings (compare Fig. 7a,b), we send where a rule $X \to Y$ approximately holds according to some parameter $k$, which defines the number of items of the tail that have to be present in the transaction. The errors made by this approximation are then accounted for by sending error correcting matrices $\mathcal{X}_{X \to Y}^-$ and $\mathcal{X}_{X \to Y}^+$, which account for the destructive, respectively additive noise in the are where the rule applies (compare Fig. 7c).

Let us first assume we are given a $k$, we will later show how we can optimize for $k$. We redefine the transaction sets $T_{Y|X} = \{t \in D \mid (X \subset t) \wedge (|Y \cap t| \geq k)\}$, which corresponds to the transactions where the rule approximately holds. We will now slightly abuse notation and indicate the binary input matrix that correspond to $D$ by $\mathcal{D}$, and we subset this matrix using the transaction id lists and item subsets. Both of these are sets of indices that indicate which rows, respectively columns to use of the matrix. For example, the submatrix where $X$ holds is given by $\mathcal{D}[T_X, X]$. We can now define the error correcting matrices to be $\mathcal{X}_{X \to Y}^- = \mathcal{D}[T_{Y|X}, Y] \otimes \mathbb{1}^{|T_{Y|X}| \times |Y|}$, and $\mathcal{X}_{X \to Y}^- = \mathcal{D}[T_X \setminus T_{Y|X}, Y]$, where $\otimes$ is the element-wise XOR operator and $\mathbb{1}^{i \times j}$ is a matrix of size $i \times j$ filled with ones. The receiver, knowing $T_X$ and $T_{Y|X}$, can then reconstruct the original data $\mathcal{D}[T_{Y|X}, Y] = \mathbb{1}^{|T_{Y|X}| \times |Y|} \otimes \mathcal{X}_{X \to Y}^-$, respectively $\mathcal{D}[T_X \setminus T_{Y|X}, Y] = \mathcal{X}_{X \to Y}^+$.

While this explains the concept of how error correcting matrices can be used to reconstruct the original input, which hence define a lossless encoding, we are mainly interested in the codelength functions. To adapt the data costs, we now additionally send the two error matrices, which we can do using binomial codes. Hence, we get

$$L(D \mid M) = \left( \sum_{X \to Y \in M} \log \binom{|T_X|}{|T_{Y|X}|} \right) + \left( \sum_{I \in \mathcal{I}} \log \binom{|D|}{|T_I'|} \right)$$

$$+ \log \binom{|T_{Y|X}| \times |Y|}{|\mathcal{X}_{X \to Y}^-|} + \log \binom{|T_X \setminus T_{Y|X}| \times |Y|}{|\mathcal{X}_{X \to Y}^+|},$$

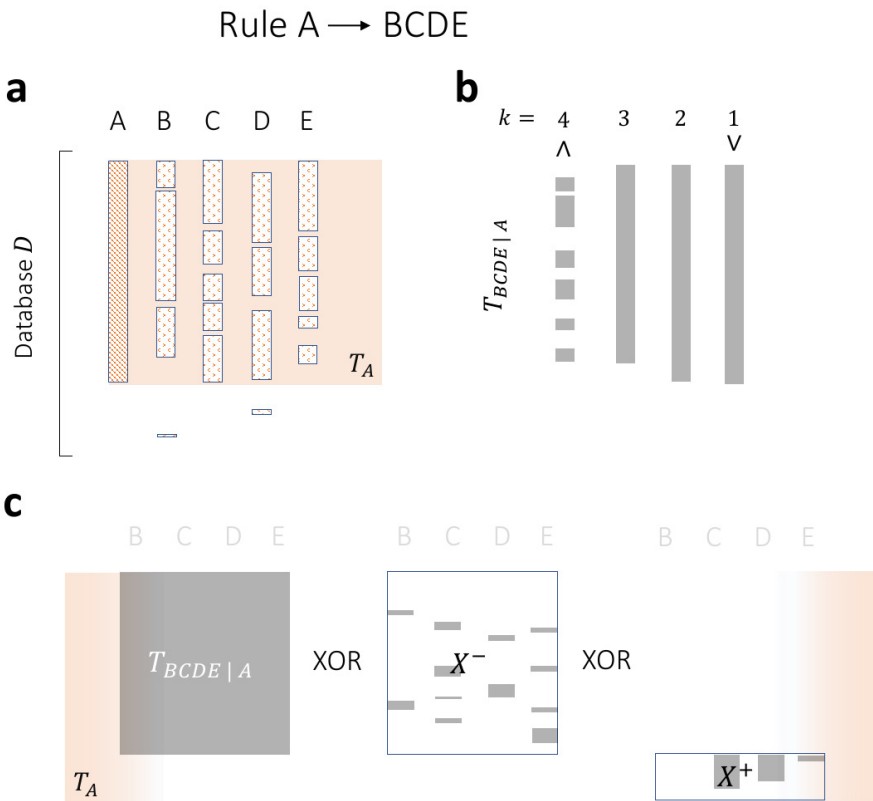

Figure 7: *Example of tail error encoding.* For a given database $D$ given in **a**, where blocks indicate the occurrence of items, a good rule is given by $A \rightarrow BCDE$. The part of the database where the rule applies is indicated by the orange area. In **b** we show the part of the transaction were the rule holds for varying number $k$ of tail items that have to be present in a transaction, from all items on the left – corresponding to a conjunction – towards just a single item on the right, which corresponds to a disjunction. In **c** we visualize the error encoding used to transmit the data for $k = 3$. We first transmit the data where the rule holds, resulting in the area that is indicated by the gray block. XORing the error matrix $X^-$ with this block, it is possible to reconstruct the original data for the part where the rule holds. Using $X^+$, we reconstruct the original data in the area where the rule applies but does not hold.

with the second line providing the codelength of the error matrices, and $|\mathcal{X}|$ indicating the number of ones in $\mathcal{X}$.

Our model $M$ now not only consists of rules $M \cup M_{ind}$, but also of the set of error correcting matrices. As the submatrix to which we need to apply the matrix is fully defined by $T_X, T_{Y|X}$, and $Y$ of the corresponding rule, also defining its size, the only adaptation we need for the model costs is the parametric complexities induced by the codes for transmitting the data. This yields

$$L(M) = |\mathcal{I}| \times L_{pc}(|D|) + \sum_{X \rightarrow Y \in M} L(X \rightarrow Y) + L_{pc}(|T_{Y|X}| \times |Y|) + L_{pc}(|T_X \setminus T_{Y|X}| \times |Y|).$$

This completes the MDL costs for rules robust to noise in the tail for a given $k$. To optimize $k$, the crucial insight is that the codelength of individual complex rules are independent, as is the data cost. That means we can optimize a $k$ for each rule separately. Thus, for a given rule $X \rightarrow Y$ we can enumerate all $|Y|$ many models for the different thresholds $k$ and let MDL decide which one fits the data best.

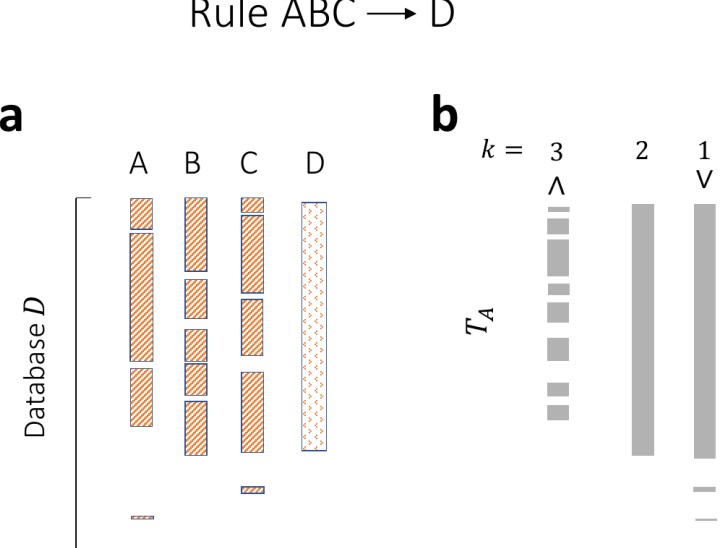

Figure 8: *Example of the impact of noise.* For a given database $D$ given in **a**, where blocks indicate the occurrence of items, a good rule is given by $ABC \rightarrow D$. Due to high noise, the simple conjunctive pattern language results in a bad representation on where the rule should apply, visualized on the left of **b**. More relaxed definitions towards disjunctions, where we only require $l$ items of the head to be present in the transaction, result in much more stable representation on where the rule applies.

### A.3 THE IMPACT OF THE EXTENDED PATTERN LANGUAGE

Extending the pattern language for rule heads is crucial to be applicable for tracing activation patterns through a neural network. First of all, we need to start from labels, which are inherently activated mutually exclusive, as we only have a single label as classification. To find shared features of labels, it is essential to be able to express disjunctions with rule heads. Furthermore, the data as well as activation patterns across the data are very noisy. Thus, determining where a rule applies just based on conjunctions of features can give a very twisted look of the data at hand, as visualized in Fig. 8. That is the reason to introduce a more flexible language similar to approximate rule tails, which solves these issues.

### A.4 SEARCH COMPLEXITY

The size of the search space implied by our model class $\mathcal{M}$ is in $O(2^{|I_i| \times |I_j|} \times 2^{|I_i| + |I_j|})$. For two layers $I_i, I_j$, we enumerate all possible rules by

$$\underbrace{\left(\sum_{k=0}^{|I_i|} k \times \binom{|I_i|}{k}\right)}_{\text{Possibilities for head}} \times \underbrace{\left(\sum_{l=0}^{|I_j|} l \times \binom{|I_j|}{l}\right)}_{\text{Possibilities for tail}}$$

$$\leq |I_i| \left(\sum_{k=0}^{|I_i|} \binom{|I_i|}{k}\right) \times |I_j| \left(\sum_{l=0}^{|I_j|} \binom{|I_j|}{l}\right)$$

$$= |I_i|\, 2^{|I_i|} \times |I_j|\, 2^{|I_j|} = |I_j|\, |I_i|\, 2^{|I_i| + |I_j|},$$

where the first sum enumerates all heads of size $k$, the binomial coefficient describes the ways of drawing heads of such size, and the term $k$ is the number of models given by the robust head encoding. Similarly, the second sum enumerates all tails of size $l$, the binomial coefficient describes the drawing of such tails, and the term $l$ is the number of ways to place the error correcting matrices

for the robust tail encoding. As in theory we can have any subset of these rules as a model, we thus get approximately $2^{(|I_j| \times |I_i| \times 2^{|I_i|+|I_j|})}$ many different models.

## A.5 ALGORITHM PSEUDOCODE

EXPLAINN explores the search space of rule sets in an iterative fashion, either generating new rules with a single item in the tail, or merging two existing rules, thus generating more complex rules with multiple items in the tail. Using these two steps, we can generate all potential candidate rules to add to the model, and evaluate their respective gain in terms of MDL. For a rule $r'$, we will say model $M' = M \oplus r'$ is the refined model, with the refinement operator $\oplus$ adding the rule $r' = X \to Y$ to $M$, removing the merged rules that led to $r'$, if any, and updating the singleton transaction lists $T_A$ for all items in the tail $A \in Y$. Here, we will provide the pseudocode for the two candidate generation functions for new rules and for merging rules in the general setting alongside the complete algorithm of EXPLAINN.

For generating a new rule with a head using the extended pattern language we use the approach described in the main paper, gathering all confidence values for a given neuron $A$ in $I_j$ for all potential head neurons $I_i$. We keep all potential head neurons with confidence value beyond $\theta$ in a list $H_A$ sorted descending on confidence and merge the first $t$ neurons in the list to form the head. Going over all $t = 1..|H_A|$ allows us to greedily optimize for the best of all relevant heads for the given item,.

---

**Algorithm 1:** GENCANDIDATENEW

**Input:** Dataset $D$ over layers $I_i, I_j$, Model $M$, tail item $A$, threshold $\theta$
**Output:** Best refinement $M'$

1  $H_A \leftarrow \emptyset$ ;                                                    // head items, in decreasing order of confidence
2  **foreach** $x \in I_i$ **do**
3     $\quad \sigma_{x,A} \leftarrow \frac{|T_x \cap T_A|}{|T_x|}$;               // Compute conditional frequency
4     $\quad$ **if** $\sigma_{x,A} > \theta$ **then**
5        $\quad\quad$ **insert** $(x, \sigma_{x,A})$ **into** $H_A$;              // Add neuron $x$ to list

6  $M' \leftarrow$ **none**;
7  $\Delta_{min} \leftarrow 0$;                                                   // gain estimate in bits
8  **for** $t = 1...|H_A|$ **do**
9     $\quad M' = M \oplus \{H_A[: t] \to A\}$;                                   // Refine model with rule using first $t$ labels
10    $\quad \Delta_t \leftarrow L(D, M) - L(D, M')$;
11    $\quad$ **if** $\Delta_t < \Delta_{min}$ **then**
12       $\quad\quad \Delta_{min} \leftarrow \Delta_t$;
13       $\quad\quad M' \leftarrow M \oplus \{H_A[: t] \to A\}$;                  // Update best rule set

14 **return** $M'$

---

The key component is hidden in the gain estimate in line 10, which for the given rule $X \to A$ determines the best value $k$ of items in the head needed for a rule to apply. That is, we test all for all transactions sets determining where the rule applies $T_X^k = \{t \in D \mid |X \cup t| \geq k\}$ which one gives the best gain. To generate new rules going from the output layer to a hidden layer, we want to mine rules with disjunctive heads, which means we only have to consider $T_X^1$ – corresponding to a disjunction – in the search process.

To generate candidates from existing rules in $M$, we use an extended search scheme that allows to merge pairs of rules with approximately equal heads, hav-

ing up to $\mu$ dissimilar items, measured by the symmetric set differences between $\ominus$.

---

**Algorithm 2:** GENCANDIDATEMERGES

---

**Input:** Dataset $D$, Model $M$, overlap threshold $\mu$
**Output:** Candidates $C$ sorted by gain $\Delta$
1  $C \leftarrow \emptyset$ ;                                                                   // Candidate rule merges
2  **for** $r_1 = X_1 \rightarrow Y_1 \in M, r_2 = X_2 \rightarrow Y_2 \in M$ **do**
3     **if** $|X_1 \ominus X_2| \leq \mu$ **then**
4          $\Delta_\cap \leftarrow L(D, M \oplus \{X_1 \cap X_2 \rightarrow Y_1 \cup Y_2\}) - L(D, M)$;   // Gain of adding conjunction of heads
5          **if** $\Delta_\cap < 0$ **then**
6              **insert** $(X_1 \cap X_2 \rightarrow Y_1 \cup Y_2,\ \Delta_\cap)$ **into** $C$ ;            // Add to candidates
7          $\Delta_\cup \leftarrow L(D, M \oplus \{X_1 \cup X_2 \rightarrow Y_1 \cup Y_2\}) - L(D, M)$;   // Gain of adding disjunction of heads
8          **if** $\Delta_\cup < 0$ **then**
9              **insert** $(X_1 \cup X_2 \rightarrow Y_1 \cup Y_2,\ \Delta_\cup)$ **into** $C$;            // Add to candidates

10 **return** $C$

---

Using the candidate generation methods, we can now write down the pseudocode for EXPLAINN, which iteratively generates candidates and commits to the candidate with highest gain, until there is no more candidate that yields any gain in terms of MDL.

---

**Algorithm 3:** EXPLAINN

---

**Input:** Dataset $D$ over layers $I_i, I_j$, frequency threshold $\theta$, overlap threshold $\mu$
**Output:** Best model $M^*$
1  $M \leftarrow \{\emptyset \rightarrow A \mid A \in I_j\}$ ;                            // Initialize model with baseline rule set
2  **foreach** $A \in I_j$ **do**
3     $R' \leftarrow$ GENCANDIDATENEW$(D, M, A, \theta)$;                        // App. Alg. 1
4     $M' \leftarrow M \oplus R'$;
5     **if** $L(D, M') < L(D, M)$ **then**
6          $M \leftarrow M'$;

7  **repeat**
8     $\hat{M} \leftarrow M$;
9     $C \leftarrow$ GENCANDIDATEMERGES$(D, M, \mu)$;                    // App. Alg. 2
10    $\mathcal{Y} \leftarrow \emptyset$;                        // Keep track of independence of merged rules
11    **foreach** $X \rightarrow Y \in C.\, Y \not\subset \mathcal{Y}$ **do**
12         $M' \leftarrow M \oplus \{X \rightarrow Y\}$;                  // Refine model, test gain
13         **if** $L(D, M') < L(D, M)$ **then**
14              $\hat{M} \leftarrow M'; \mathcal{Y} \leftarrow Y$;

15 **until** $M = \hat{M}$;
16 **return** $M$

---

# B    EXPERIMENTS AND DATA

Here, we detail the setup and training of the individual networks, and provide further experimental results.

## B.1    MNIST TRAINING

We trained a CNN on the MNIST data set using the Keras framework, using 60000 images for training and 10000 images as hold out test set for evaluation. The network consists of 2 convolutional layers, with 20 filters in the first layer and 40 filters in the second layer, each using 3x3 kernels and 2x2 maxpooling. The convolutional layers are followed by a Dropout layer with dropout rate .25, and the flattened outputs are passed on to a fully connected layer with 64 nodes with *ReLU* activations. Then follows a dropout layer with rate .5 and the output layer of size 10 with softmax activations. The network was trained using AdaDelta with default parameters based on categorical cross entropy loss over 12 epochs using a batch size of 128. We gathered binarized activations across all filters and applied EXPLAINN to build rules from the output layer to the first respectively second convolutional layer.

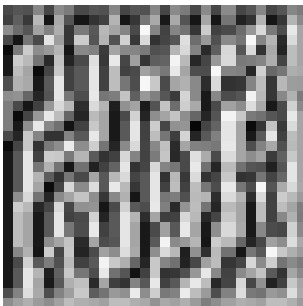

Figure 9: *MNIST prototype.* Prototype image for filter 2 from the first convolutional layer.

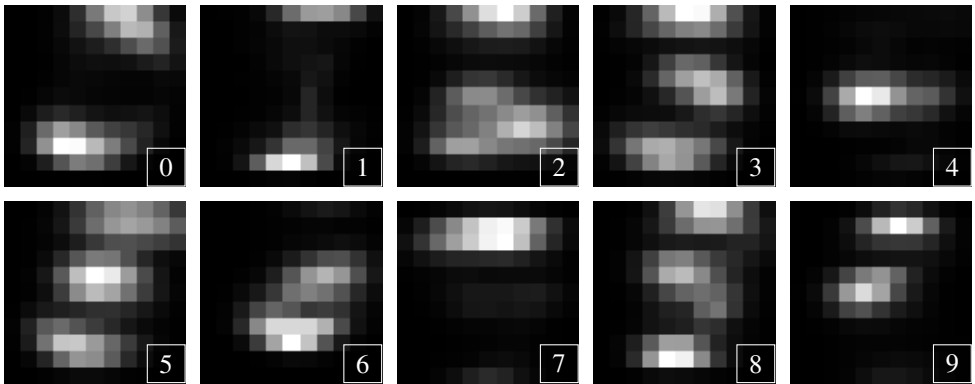

(a) *MNIST Average activations.* Average activation maps across a class for filter 32.

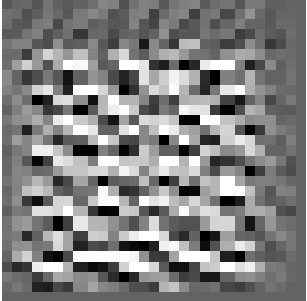

(b) *MNIST Prototype.* Prototype image for filter 32.

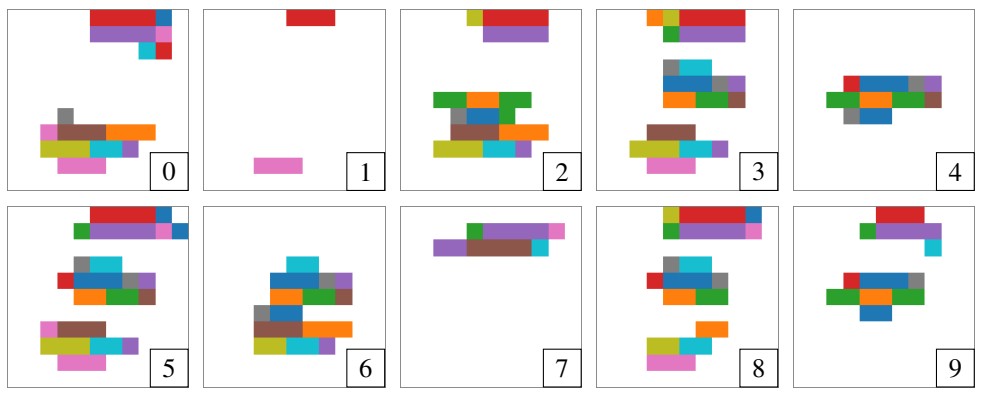

(c) *Horizontal edge detector.* Discovered rules, feature groups found across classes share the same colour.

Figure 10: *Filter visualizations.* Activation maps (a) for the classes, the prototype of the filter (b), and discovered rules (c), over the whole dataset for filter 36 in the second convolutional layer.

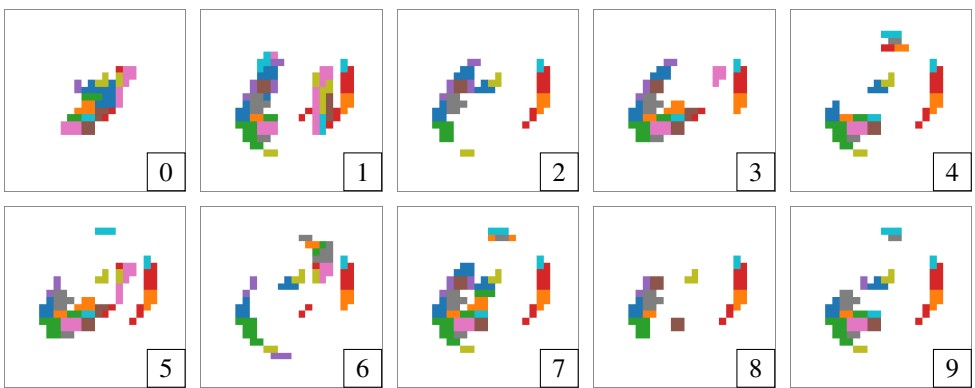

Figure 11: *The negative of a digit.* Visualizations for filter 12 in the first convolutional layer. This filter seems to capture the 'negatives' of the handwritten digits.

## B.2 IMAGENET FURTHER RESULTS

### B.2.1 GOOGLENET RESULTS

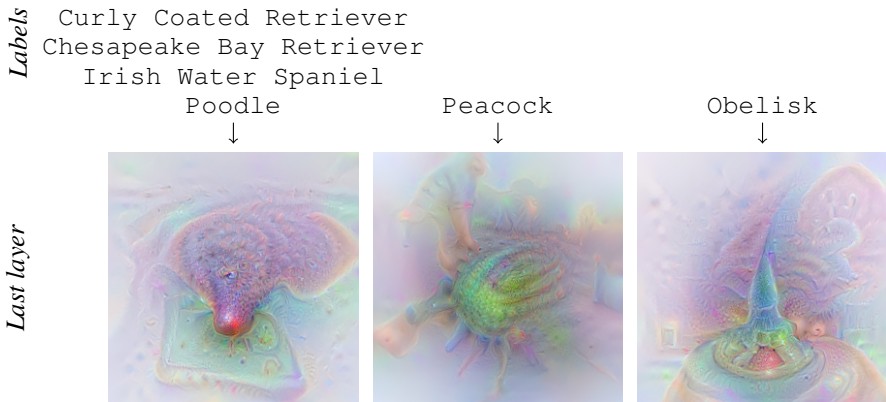

(a) Prototypes for rules from output label to last hidden layer.

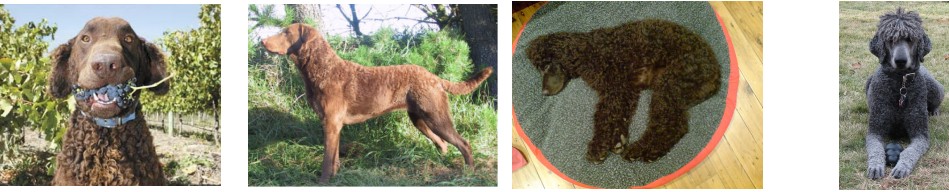

(b) Samples for curly haired dog breeds. From left: `Curly Coated Retriever`, `Chesapeake Bay Retriever`, `Irish Water Spaniel`, `Poodle`

Figure 12: *GoogLeNet results on ImageNet.* (a)Visualizations for the rules found between the labels and the last hidden layer in GoogLeNet. The labels in the rule heads are written above the prototype images of the tail unit groups. Each rule tail captures some interesting features of the corresponding classes: In the first rule the characteristic curly hair of different dog breeds is captured, the second group encapsulates information about the typical colourful plumage of peacocks, the third captures the shape of obelisks. We provide example images of the curly haired dog breeds in (b).

### B.2.2 VGG SHARED NEURONS

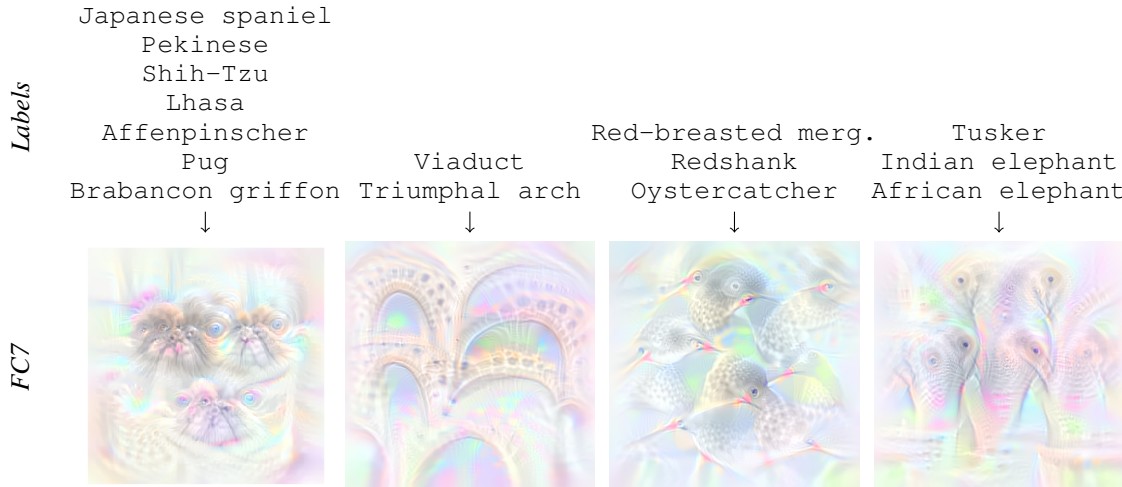

Figure 13: *Shared information across labels.* Visualizations for the rules found between the labels and the last fully connected layer (FC7). The labels in the rule heads are written above the prototype images of the tail unit groups. Each rule tail captures some interesting features of the corresponding classes: In the first rule the characteristic face of different dog breeds is captured, the second group encodes information about the arch structures present for both `Viaduct` and `Triumphal arch`, the third captures the red beaks surrounded by blackish feather that are shared between different birds, and the fourth shows typical heads and tusks of elephants.

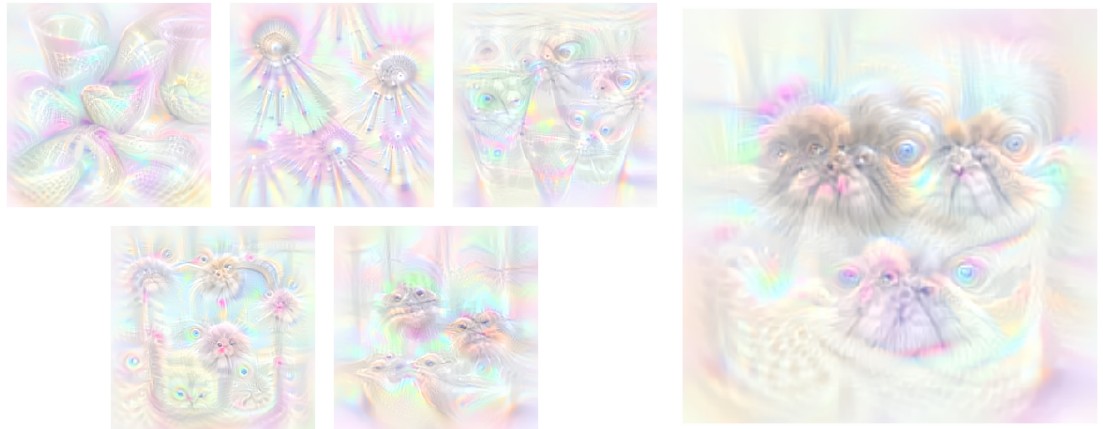

Visualization for the units in the tail individually      Visualization for the whole tail

Figure 14: *The ugly face neurons.* From the data for all dog breed categories, EXPLAINN discovered the rule between the labels {`Japanese spaniel, Pekinese, Shih-Tzu, Lhasa, Affenpinscher, Pug, Brabancon griffon`}, and 5 units from the *FC7* layer, for which a prototype is given in the right image. The 5 units together capture the rather characteristic face of the mentioned breeds. Visualizing these 5 units one by one, given on the left, gives only little insight about the encapsulated information

We discovered many shared traits that the networks is able to pick up across classes, which are encapsulated in groups of neurons in the last layer. For example, there are neurons that capture the red beaks of different birds, arch like structures of buildings, tusks of elephants and as an object itself, and the ugly face of a whole group of different dog breeds (see App. Fig. 13). Even if we would visualize all neurons individually and handpick them, without the knowledge about the rule,

e.g. for the ugly dogs, it is very hard to interpret what they encode, and only few of them give hints about what the information is (see App. Fig. 14). In practice, we only visualize class prototypes, which can be very misleading, as shown in the next section.

### B.2.3 RESOLVING THE MEANING OF CLASS PROTOTYPES

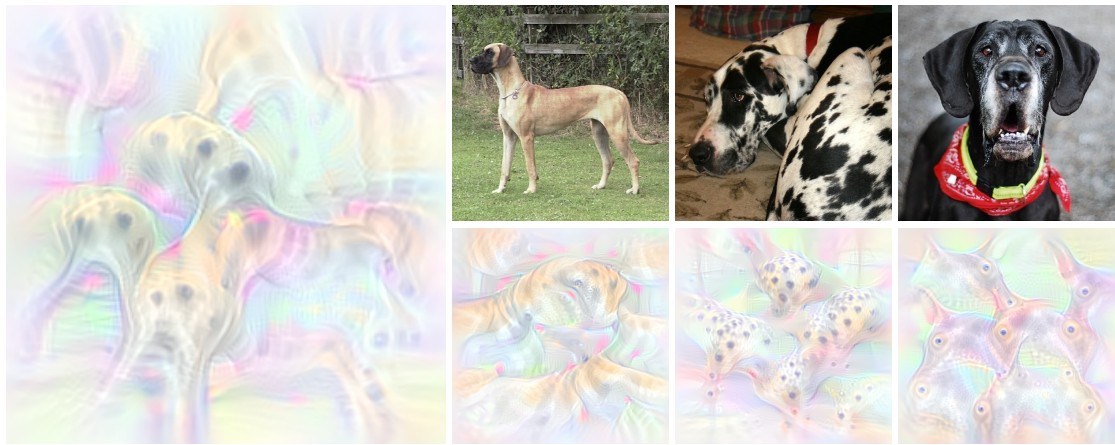

Class prototype for `Great Danes`                    Row 1: Example images from the class
                                                     Row 2: Rules in association with `Great Danes`

Figure 15: The left image shows the visualization for the whole class `Great Danes`. This visualization could not highlight many characteristic features, since there is a large diversity within the class. On the right side 3 images from the dataset is shown, along with 3 rules that EXPLAINN finds in connection with the class label. We are able to pick up trends, that are not characteristic to the whole class, but only a subset.

A standard technique to capture traits that a neural network learns about a class is prototyping the given class label. These can offer hints on what the network learns globally about the class, but very often lead to uninterpretable results. We provide one such example prototype for the `Great Dane` class in Imagenet of the VGG network in App. Fig. 15, which does not provide any clue what the network learns. The rules discovered by EXPLAINN however show, that different groups of neurons in the last layer lead to the `Great Dane` classification, each encoding a distinct type of fur colour and pattern that appear with this breed. The class prototype is a mixture of these different types, which explains the difficulty to interpret that prototype.

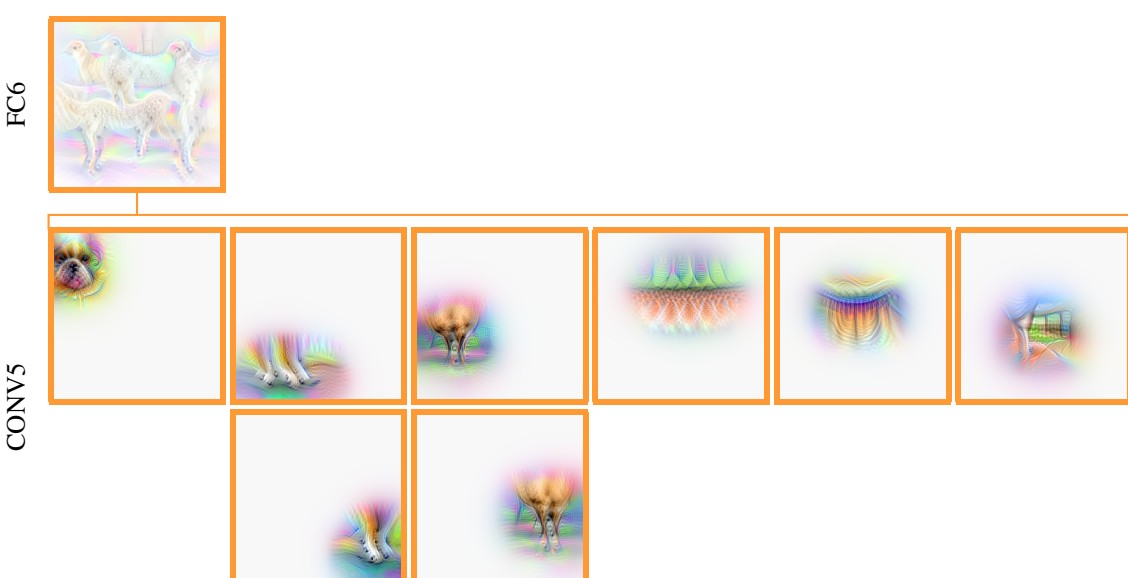

Figure 16: *Side dogs*. The visualization for the *FC6* nodes resembles animals from side view, a typical image positioning for dogs. In the images generated for associated units in the convolutional layer, we see how the network uses different filters to put this pattern together. The first depicted filter captures a dog head, typically found at the top left part of the image. The second and third filters focus on the legs. In the third row, we can see that other units for these filters are also commonly activated with the feature, showing that there are typical positions for legs in both the left and right lower parts of the image. In the last three prototypes, we recognize horizontal lines, which overall resembles the typical shape and position in the image where we would expect the back of an animal.

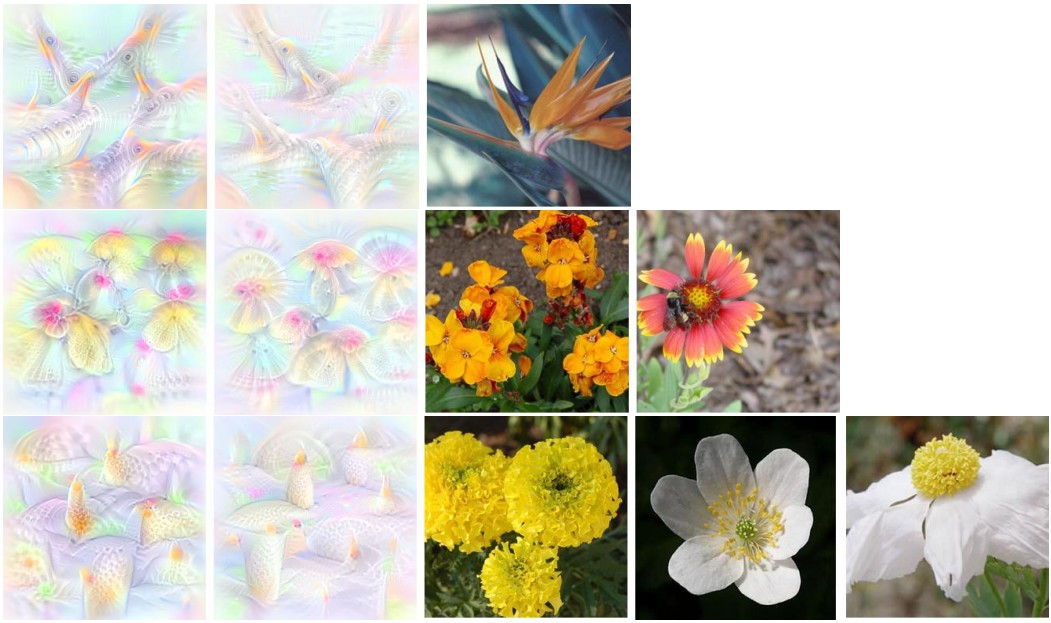

Figure 17: *Flower visualizations.* For rules found between output and last fully connected layer, we visualize the neurons in the tail of the rule for the fine-tuned VGG-S network (first), the original VGG-S network (second), and example images for the flower classes (right).

## B.3 OXFORD FLOWER DATA

## B.4    ADDITIONAL RESULTS

In App. Fig. 18, 19, we provide additional results based on prototyping for rules found for *ImageNet* data and the VGG-S network. We focus on rules with multiple neurons in the tail, as such class and multiclass prototypes can hardly be found by hand. Overall, we observed that the larger the number of neurons in the tail, the sharper and more interesting the resulting protoype. Furthermore, we found that for many prototypes spanning multiple classes, we discover multiple rules for some of these classes (e.g. `Black Grouse`) and the protoypes indicate that only a fraction of information, such as patterns, a colored leg or beak, or a color patch, is used from each group of neurons such that together they arrive at the class prediction.

In App. Fig. 18, the first row of the panel are examples of neuron groups that learn typical shapes of objects, such as `Sombrero` or `Gondola`. The second row contain groups of neurons capturing typical patterns and colors for individual classes, such as yellow patches on black skin of the `Fire Salamander`, red caps with white dots of the `Agaric` mushroom, the typical leaf with red veins of `Sorrel` or the wings of a `Monarch` butterfly. The third row contains common features between two classes that are together captured by the same group of neurons, like the arch-like structures and round rooftops found for certain `Triumphal Archs` and `Mosques`, the layered and intertwined worm-like shapes of many `Fur Coats` and the `Gyromira` mushroom, or the characteristic traditional covering of yurts and the front part of dogsleds.

In App. Fig. 19, groups of neurons that are shared between multiple classes are visualized both revealing surprising similarities, as well as confirming that the network learns similarities that we also use as a human. In the first row, the neurons described by the first two images capture the typical shape and red color of the ears shared between the `Red Fox` and the `Lesser Panda`, respectively the insect legs and shiny turquoise color of the body of `Tiger Beetles` and `Damselflies`. Intriguingly, the network also learns a roundish shape and distinct pattern between the `Jackfruit` and the `Squirrel Monkey`. At this point, we would like to invite the reader to look up how the top of the head of such a monkey looks like, it resembles surprisingly well the size, color, shape, and texture of a Jackfruit. For the last picture in the first row of this panel, we see dotted wings that clearly are related with the associated labels `Cabbage Butterfly`, and `Sulphut Butterfly`. But opposed to visualizations related to other Butterflies (given in both panels), the wings are all oriented in a distinct way, which resemble the cap of a dotted mushroom, which might explain the association with the `Agaric` mushroom. In the second row, we observe that the network captures common features shared between similar classes - in this case closely related animals - with the same set of neurons, which matches human intuition.

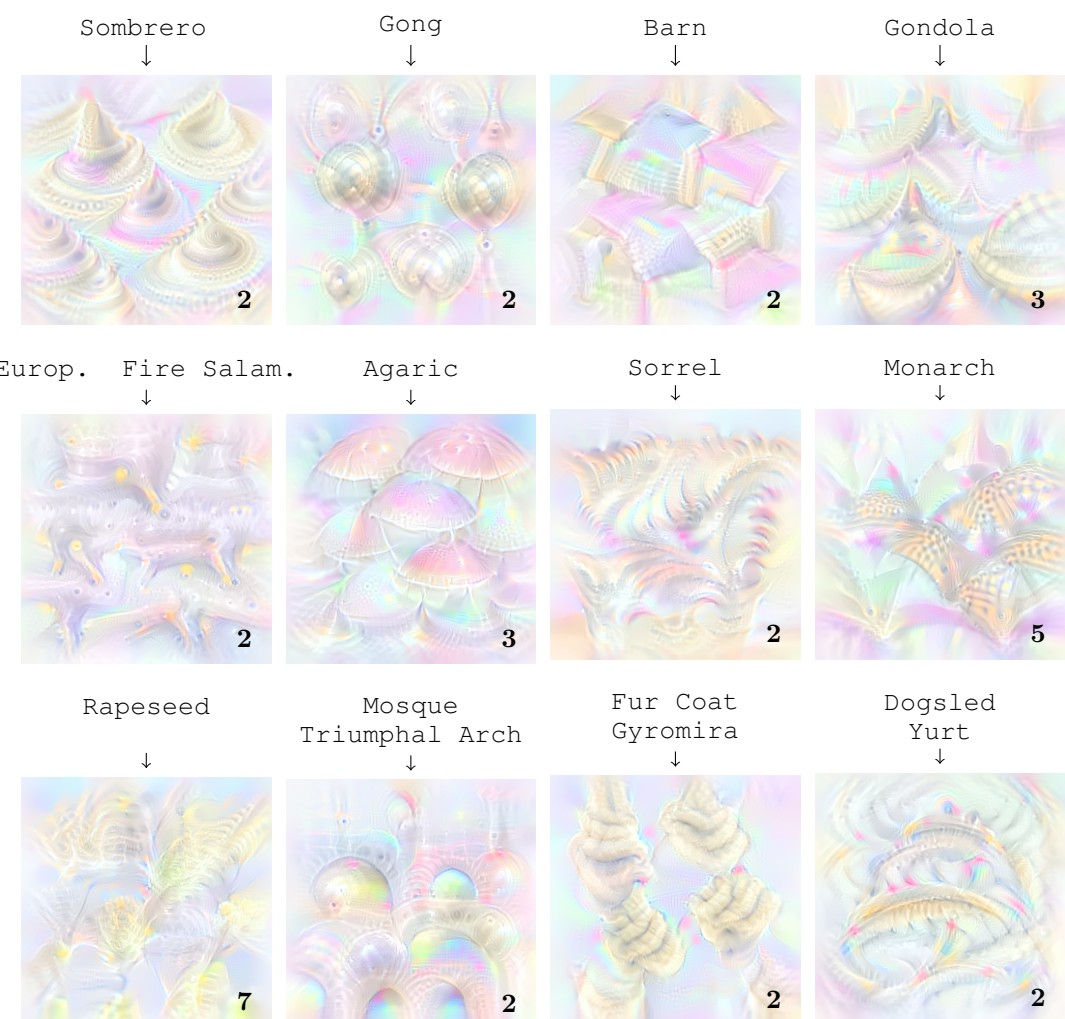

Figure 18: *Diverse prototypes*. Visualized are prototypes for rules found in the VGG-S network for *ImageNet* data between the output and last hidden layer. The class labels corresponding to the output are given above each image, the size of the group of neurons that this picture was generated from is given in the bottom right.

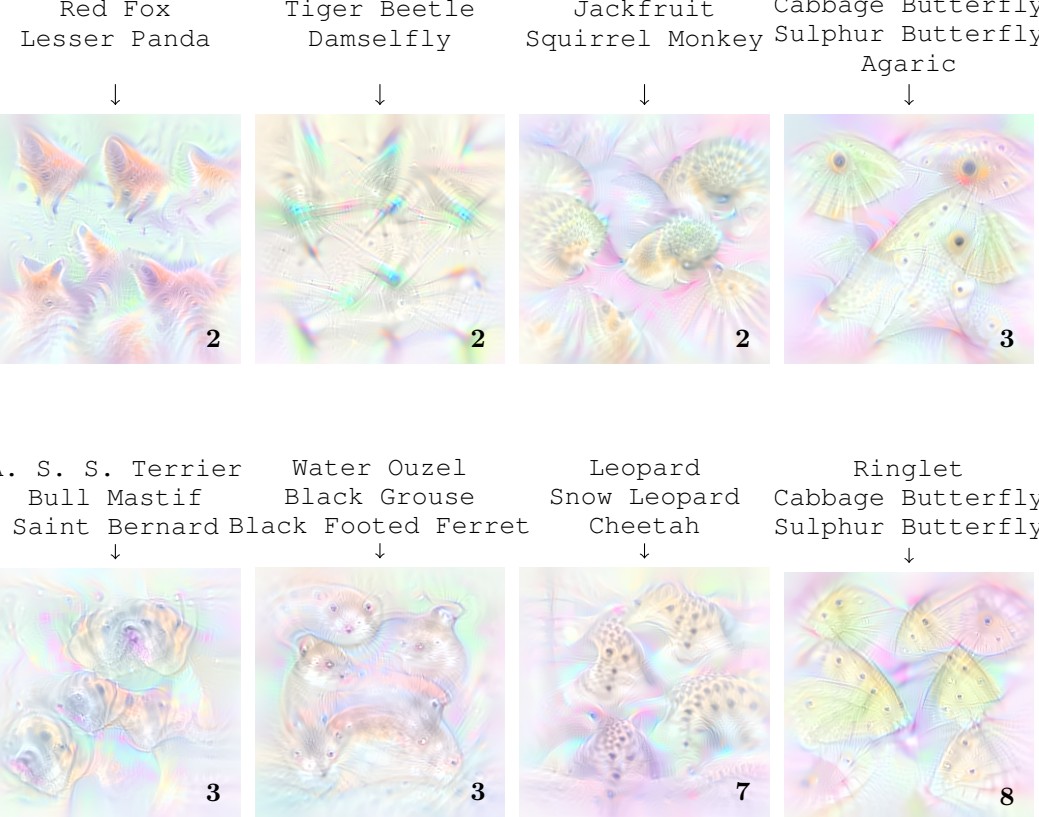

Figure 19: *More prototypes.* Visualized are prototypes for rules found in the VGG-S network for *ImageNet* data between the output and last hidden layer. The class labels corresponding to the output are given above each image, the size of the group of neurons that this picture was generated from is given in the bottom right.

