# OpenReview forum: "What's in the Box? Exploring the Inner Life of Neural Networks with Robust Rules"
_ICLR.cc/2021/Conference — Reject_

### Official Review · AnonReviewer3 · 2020-10-27
**Important complement to existing NN interpretation tools, solid implementation, some questions about usability and replicability**

**Rating:** 8
**Confidence:** 3

**Review:**

This work makes a convincing case that we need to trace information flow (or at least, to find clusters of neurons working together) within a deep neural network in order to get the clearest picture of how the network is working.  The approach is quite simple but also quite novel in that it uses concepts from coding theory which are not widely known in the deep learning community.  The paper is well written and presents examples from deep CNNs (~20 layers) trained on ImageNet.

Quality:
The method is well thought-out and explained, and two different evaluations are used (MNIST and ImageNet).  It would be more impressive if the authors had included a problem from a different data modality since in principle the method is quite general.  The examples look very impressive, but my main concern is with whether the examples could have been cherry-picked, in the sense that most of the thousands of rules produced may not be useful.  Relatedly, I would like to know the reproducibility of the result.  If you train twice with different seeds, how similar are the results?  Or if you fit ExplainNN with different seeds on the same network?  And what is the danger of false positive findings?

Clarity:
The paper is very clear with regards to the problem setting, previous work, and the methods.  I am somewhat familiar with coding theory, having read much of MacKay's Information Theory, Inference, and Learning Algorithms, but I am by no means an expert in coding theory.  Still, I am confident that I understand the principles of the method.  However, I did not exactly follow how the authors carried out the tracing (pg 7 second paragraph.). Do you just apply ExplainN as usual and then filter for rules that (strictly/non-strictly) include Y?  This seems important to explain since the tracing, in my view, is the main contribution, given that there already exist tools for understanding the similarities of classes such as representational similarity (https://roberttlange.github.io/posts/2019/06/blog-post-3/).

Originality:
I have not previously seen the idea of mining association rules for deep neural networks, although it is a simple enough idea and I would not be surprised if the idea has appeared before.  However, the application of MDL to solving the problem for binarized activations is likely to be novel.

Significance:
The work is promising, but to me it is not conclusive that it will make a lasting impact.  There are a number of important practical questions to be addressed that could make or break the method as a tool for the field, such as the reproducibility and usability of the method (whether most rules produced are meaningful or significant manual filtering is required).  On the other hand, even if the method does not meet practical needs, it could still be of great utility for NN methods researchers interested in investigating, say, the redundancy of specific architectures. the true degree of similarity between two trained models, convergent dynamics between alternative architectures, and the value of overparameterization.  The paper could be even more significant if the authors could comment on the generalizability to other architectures such as RNNs, GNNs or transformers.  The method itself is interesting enough and the examples sufficiently compelling (even if cherry-picked) that I would recommend the paper to almost anyone interested in neural network interpretability.

Note on rating:  In the face of limited details, I am willing to give the authors the benefit of the doubt that they did not cherry-pick overly aggressively and that the examples are representative of typical outputs. If it turns out that most rules do not look like the examples, then my rating would decrease.  On the other hand, if the authors can address my concerns about cherry-picking in the response, it is possible that I would raise my rating.

Pros:
 * important application
 * method is quite general
 * method is simple and intuitive

Cons:
 * evaluation of method performance limited to selected examples
 * reproducibility not addressed
 * control of false positives not addressed
 * method for tracking across layers not well-explained

---

> ### Author Response · Authors · 2020-11-13
> **Response to Reviewer 3**
>
> Thank you for this extensive and positive feedback.
>
> - If you train twice with different seeds, how similar are the results?
> This is a very interesting question. We have not yet considered this, but indeed, it should be possible to use ExplaiNN as a tool to see if two networks (e.g. trained on the same data, but with different seed) have learned the same main concepts! Essentially, if the networks have learned the same concepts, have approximatey the same generalization error, then between the networks the neurons are merely "permuted" and hence ExplaiNN would discover the same rule sets for both networks (up to neuron ids). The moment that ExplaiNN discovers strongly diverging rule sets, this gives evidence that the networks learned intrinsically different concepts. We could even consider how well the rule set for Network A can be used to compress (after neuron-id optimization) the activations of Network B (and vice-versa). Far beyond the scope of this paper, but fascinating!
>
> - If you fit ExplaiNN with different seeds on the same network?
> Maybe we misunderstood the question, but ExplaiNN is deterministic, for the same data iw will always return the exact same result.
>
> - What is the danger of false positive findings?
> If we interpret this question correctly, a false positive finding corresponds to a rule that does not reflect information that is important for the data. If that is the case, it is also very unlikely that these neurons are sufficiently often co-activated for a rule over them to improve compression compared to the base-line model, which assumes independence between neurons. In a nutshell, MDL ensures that only rules for which sufficient evidences exists in the data are added to the model, as otherwise the number of bits needed to encode data and model would unnecessarily increase. This question will also be implicitly covered by the new synthetic data experiments in response to Reviewer 1.
>
> - Question about tracing.
> That is a good point, we will add a more extensive description in the revised manuscript. Suppose we have three consecutive layers L1 -> L2 -> L3. We first discover all rules X->Y, X \subset L3, Y \subset L2. In the next step, for each such Y, we run ExplaiNN to discover rules Y->Z, where Z \subset L1. There also exist other approaches that would work, for example mining rules between each consecutive layer pair, and then combine those that "overlap" -- which we think is what your comment described. Our iterative approach has the advantage that we can force the model to focus on what is necessary to build traces and thus get cleaner results, with the general pairwise application we have to do the filtering by overlap that you mentioned to get the clean results.
>
> - Concern about cherry-picking
> We could not agree more with this concern -- not so much for ExplaiNN, but for explanatory approaches for CNNs in general. A key  advantage of our MDL-based approach is that we discover succinct and non-redundant sets of rules, here resulting in hundreds to at most a few thousand rules, which we can analyze in its entirety. For the VGG-S network, the existing highly optimized prototyping allowed us to generate prototypes for all discovered rules. For the large majority the result looks meaningful,  it is clear what information they encode, and how it relates to the output. As common with prototyping, there are also few that are hard to interpret, or reveal insights only on second glance, because they e.g. encode abstract information (geometric shapes, small subparts of an object). For rules with larger heads (spanning many classes) the prototypes often reflect some of the classes well, while the other classes seem to utilize only certain information from the neuron (e.g. a pattern, a red leg). Such classes are classified by information of multiple neuron groups - found in several rules - which together provide the whole picture. Overall we picked a representative subset that reflects the information content of most of the prototypes. We will make the code available once the paper is published, such that the results can be reproduced and everybody can take a look on what the network sees. We now also added two additional panels containing 20 additional prototypes to the appendix, which we think are larger groups of representatives of our findings.

---

> > ### Comment · AnonReviewer3 · 2020-11-20
> > **Some suggestions**
> >
> > > Thank you for this extensive and positive feedback.
> >
> > You are welcome.  And this is more directed to the other reviewers, but the reason for my high score is my own intuition that this method has a special quality of being very "elegant" and "practical" that comes from my intuition from developing machine learning methods for several years.  This intuition is hard to defend, but nevertheless it gives me extra confidence in this method.  I admit that I am not exhaustively familiar with other approaches in the literature, but I have seen enough that this work stands out from the typical work I have seen on NN interpretation.  But this does not at all negate the very valid point the other reviewers make which is that comparison with other works, to the extent possible, would greatly increase the value of this paper.
> >
> > > What is the danger of false positive findings? If we interpret this question correctly, a false positive finding corresponds to a rule that does not reflect information that is important for the data. If that is the case, it is also very unlikely that these neurons are sufficiently often co-activated for a rule over them to improve compression compared to the base-line model, which assumes independence between neurons. In a nutshell, MDL ensures that only rules for which sufficient evidences exists in the data are added to the model, as otherwise the number of bits needed to encode data and model would unnecessarily increase. This question will also be implicitly covered by the new synthetic data experiments in response to Reviewer 1.
> >
> > My background is in statistics, and statisticians do not trust that a method can control false positives in the absence of strong theory that defines the data assumptions, the criteria for false positive control (e.g. control of worst-case probability of type I error for any hypothesis, AKA family-wise error control) and a theorem or at least a compelling theoretical argument that the criteria for error control is satisfied under the stipulated conditions.  This is a pragmatic attitude that the statistics community has adopted over decades of scientific data analysis and dealing with all kinds of failure modes.  I think it is a mindset that the CS and AI communities will grow to adopt as well, as researchers start to deal with the difficulties of deploying AI in the public sphere--no doubt you have already seen the increased focus on interpretability and robust AI in recent years.
> >
> > From this point of view, I would have to see theoretical work establishing the validity of MDL as a form of statistical inference before I could trust it.  Are you aware of such work?  If so, it would make sense to cite it in your main paper.  And even regardless of theory, statisticians always like to see empirical demonstration of error control as well.  Hopefully your simulations will shed light on this.
> >
> > > Overall we picked a representative subset that reflects the information content of most of the prototypes. We will make the code available once the paper is published, such that the results can be reproduced and everybody can take a look on what the network sees. We now also added two additional panels containing 20 additional prototypes to the appendix, which we think are larger groups of representatives of our findings.
> >
> > Thanks for claiming that you tried to avoid cherry-picking.  However if I may make a suggestion, it is that you should be aware of unconscious biases that could lead even well-intentioned researchers to skew their results unknowingly.  Therefore, it makes sense to intentionally plan your experiments (e.g. having a completely safe test set) and the presentation of your results in such a way that the audience of your paper only has to trust you on the basis of not making outright lies in order to trust your results.  To make it concrete, you should promise to **randomly** select the examples that you chose to present in the main paper or appendix so that absolutely no human judgment goes into selection of the examples.  That would be the more convincing to readers and reviewers than choosing 20 examples and promising that these were "representative."

---

> > > ### Author Response · Authors · 2020-11-24
> > > **Comment about MDL as statistical inference (Part 1)**
> > >
> > > Thank you again for this comment. The suggestion of randomly selecting examples is interesting, however, there will always be the doubt by critical reviewers that the random selection was again biased or it was not executed faithfully. One of the main problem is also that we might completely miss on the diversity of the results when just picking a small subset. We hope that by publishing the code, which runs fast and without need of expensive hardware, people can simply reproduce the results and see on their own. Furthermore, with the new experiments thanks to the great suggestions of reviewer #1 and #4, we were able to assess quantitatively how good the rules set is as a whole -- with a synthetic data set with known ground truth, and by doing an interventional study on a real world network using classification performance as a proxy.
> > >
> > > Regarding MDL as statistical inference, we think we misunderstood your initial concern.
> > >
> > > MDL is an information theoretic model selection criterion. There exist two main variants, two-part (or crude) MDL [1] and one-part (or refined) MDL [3]. Two-part MDL can be explained both from a Bayesian perspective, as well as from an algorithmic complexity perspective. That is, in Bayesian learning we aim to maximize p(M|D) = P(D|M)P(M) / P(D) \propto P(D|M)P(M), ie. we want to maximize the conditional likelihood P(.|.) of the data given model taking into account a given prior P(.) over the model class. From Shannon information theory we know that the optimal way to encode a message is with a code of length -log P(.) bits (which does assume a distribution of messages exists). Hence, if we apply a negative logarithm to the Bayesian objective, we have -log P(M)-log P(D|M), with the first term being the number of bits to determine model M from the model class -- exactly like L(M) does in MDL -- and the second term being the number of bits to determine data D given model M -- exactly like L(D|M) does in MDL. That is, the best Bayesian model also minimizes the number of bits -- given the prior, of course. Philosophical and minor differences aside, how we treat the prior is then the main difference between Bayesian and MDL based learning. In Bayes, essentially 'any' prior that the user 'likes' goes, while in MDL we require the model to be reconstructed (penalized) from scratch, without loss, and are hence restricted to using Universal distributions [3]. Loosely speaking, in MDL we do not just penalize for the *number* of parameters in M [2], but also for the values these take [3]. [It goes much too far to here dive into here, but Refined MDL [3] also takes rewards/penalized by how easily parameters are set, i.e. if the exact value does not matter much, it's cheaper to encode]
> > >
> > > MDL can also be explained (in our view even more beautifully, because distribution free) from an algorithmic complexity perspective [3]. That is, the Kolmogorov complexity K(x) of a mathematical object x is the length |p| of the shortest program p for a given Universal Turing Machine U, such that if we run p on U it returns x and halts, i.e. U(p) = x. We can split the 'content' of this program p into meaningful structure and random noise, via Kolmogorov's structure function. We start with K(x) = K(x|S) + K(S), which holds up to a constant independent of the data. S is here a set of mathematical objects (from now on, wlog, finite binary strings) that includes at least x -- we call S a model, as it generalizes the data. K(x|S) is now the length of the shortest program that can identify x in S. K(S) is the length of the shortest program that generates S, i.e. the optimal encoded length of model S. When is S a good model for x? When x is a typical element of S, when there is no structure that sets x apart from the other elements in S, when the best thing we can do is identify x in S by means of a binary index. That is, the best model S for x is the one that minimizes K(S) + log |S|, which corresponds one to one to L(M) + L(D|M).

---

> > > > ### Author Response · Authors · 2020-11-24
> > > > **Part 2**
> > > >
> > > >
> > > > The above examples establish the firm foundations of MDL, but also show that it is p-value free: we do not use hypothesis testing inside MDL, just like Bayesian model selection [2] or Kolmogorov complexity do not. The no-hypercompression inequality [3], however, which we can here re-write as P(L(D,M*) - L(D,M) > k) <= 2^{-k} shows the relationship between compression gain and p-values: the probability that we can compress data D better than by true model M decays exponentially in the number of bits gain k. While in ExplaiNN we do not advertise this explicitly, this means we can easily set a threshold alpha to control the risk for a spurious result beyond what the model selection criteria already does, by setting a minimal gain in compression that a rule R needs to achieve before it's allowed into model M, or even to decide whether a model M sufficiently improves over the null-model M_0 (independence). In theory it is also possible to include statistical tests (eg Fisher exact, etc) to post-hoc or in-search filter results to control FPR or FWER. As efficiency of the method was a prime concern, and it's not immediately clear what test to perform (permutation testing being rather costly, and very weak unless done properly), we do not do so here. Neither does GRAB, but the comparison of GRAB [5] to methods based on statistical tests clearly show that while GRAB performs as expected/hoped, its statistical competitors fare less well; we can agree there is ample room for pattern mining methods that both perform well *and* provide such guarantees. For simple conjunctive rules, the authors of GRAB [3] provide extensive experiments that the ground truth set is recovered by the greedy algorithm, and no rules are found in noise data. Our synthetic data experiments serve the purpose of showing that this holds also for our extended rule language with disjunctive heads. With the new interventional experiment, we provide hints that the rules are also relevant in real world settings.
> > > >
> > > > So, in short, we totally agree with the importance of statistical robustness. We do not have formal guarantees on FPR or FWER, but do have strong empirical reasons to believe the results are not spurious.
> > > >
> > > >
> > > >
> > > > [1] J. Rissanen. Modeling by shortest data description. Automatica, 14(5):465–471, 1978
> > > > [2] G. Schwarz. Estimating the dimension of a model, Annals of Statistics, 6 (2):461–464, 1978
> > > > [3] Peter D. Grünwald. The Minimum Description Length Principle. MIT Press, 2007
> > > > [4] Vereshchagin & Vitányi. Kolmogorov’s Structure Functions and Model Selection. IEEE Transactions on Information Theory 50(12). IEEE, 2004.
> > > > [5] Fischer, J. and Vreeken, J. Sets of robust rules, and how to find them. In Joint European Conference on Machine Learning and Knowledge Discovery in Databases (pp. 38-54). Springer, 2019

---

### Official Review · AnonReviewer2 · 2020-10-28
**Proposes to extract rules from a neural net; Not the first time this has been proposed; Lacks comparison with prior art**

**Rating:** 3
**Confidence:** 4

**Review:**

This paper proposes to extract interpretable rules from a learned neural network. The authors claim that they are the first to propose rules connecting 1) multiple neurons together, and 2) do this at a dataset level. Their approach relies on using minimum description length and well known principles from the data mining community (e.g., downward closure lemma of apriori algorithm). The authors claim that experiments conducted on image data shows that their approach leads to more faithful, interpretable rules than other approaches such as prototyping or model distillation.

Compiling rules from neural networks has been proposed before (e.g., see "Deep Logic Networks: Inserting and Extracting Knowledge from Deep Belief Networks" by Tran and d'Avila Garcez, 2018). I think the authors need to compare against such previous approaches to quantitatively show how their work extracts better rules. Otherwise, its difficult to appreciate the value of ExplaiNN. Also, the paper doesn't say anything about how faithful the rules are to the learned neural network. I mean, it seems possible that for some input the rules could produce a different output from the neural networks. There exist other works that also try to interpret neural networks consisting of affine layers with Relu activation that guarantee consistency (see "Exact and Consistent Interpretation for Piecewise Linear Neural Networks: A Closed Form Solution" by Chu et al in KDD'18). The authors should at the very least compare and contrast with such works to highlight the pros and cons. Another undesirable property of ExplainNN seems to be that it relies on a dataset to derive its rules. Is it possible that when run with a different dataset and the same learned network, ExplaiNN would produce a different set of rules? Then how much faith do we place on ExplaiNN's output?

Writing wise, the paper is presented well enough. There's a few paragraphs in the Experiments section where the authors point repeatedly to the Appendices. In the best case this makes reading a chore. I would advise the authors to refrain from using the main body of the paper as a listing of contents and simply pointing to the appendices. The pictures in the experiments section were difficult to make out. I couldn't figure out from the image whether the husky's pointed snout had been identified as a defining feature. I would hope the authors find more compelling ways to make their point.

---

> ### Author Response · Authors · 2020-11-13
> **Response to concerns of Reviewer 2**
>
> Thank you for your comments.
>
> Both references are relevant related work, and we will add them to the manuscript. Unlike our approach, both are tailored to specific types of neural networks and do not generalize to state-of-the-art architectures such as e.g. CNNs, and use explanations that are hard to interpret - either by the type of information they retrieve or the sheer size of results returned.
>
> Concretely, Tran et al. is limited to Deep Belief networks (stacked RBMs) and optimizes for high confidence rules. Mining high confidence rules has serious practical issues – millions of highly redundant and often spurious rules are mined from small data already, even when the data contains no structure. Modern pattern (set) mining methods avoid this issue by relying on statistical tests and/or information theoretic notions.
>
> The work by Chu et al. does not suffer from the pattern explosion, but is limited to very small (<10 neurons per layer) piecewise linear neural networks (see Table 3 in their paper) as it is based on (theoretically interesting) polytope analysis. Moreover, rather than characterizing what neurons inside the network encode, it aims to explain neural activity in terms of network input (see Figure 1,5,6 in their paper). Moreover, it is much harder to interpret what the polytope actually expresses, compared to how accessible rules are.
>
> Regarding your last question, we indeed propose to analyze network behaviour based on neuron activations for a given dataset.  We consider this an advantage rather than a disadvantage, as we want to learn how the network perceives the world (the input) and how and which information it composes and combines from the input to arrive at a decision. Analyzing the network without an input is like analyzing the human brain without stimulus.
> It also opens the possibility to explore how a network’s behavior changes when we change either the input distribution (train/test, ood) or when we re-train it in a slightly different context, like in the transfer learning task.

---

> > ### Comment · AnonReviewer2 · 2020-11-18
> > **Comment on correctness of ExplaiNN?**
> >
> > Thanks for responding.
> >
> > Given that I could provide you with a couple of references that you admit is relevant, and this was just off the top of my head, would you care to comment on a quantitative comparison with the referenced approaches? I think given that previous work has already proposed to extract rules from learned neural nets, a comparison is necessary. This will aid the reader by helping clarify how to contract the various approaches for pulling out rules from neural nets. Otherwise, the reader would be confused. In case such a comparison is simply infeasible then please explain to me why that is the case.
> >
> > Also, care to comment on correctness of ExplaiNN? Chu et al reference guarantees that the generated hyperplanes are correct and complete representation of the learned neural net. Can you prove something similar wrt ExplaiNN? Recently, a lot of other surveys have also made the case for explainability techniques to provide some sort of correctness guarantee otherwise its not clear how much faith to place in the "explanations" (in whatever form they be in, rules or otherwise). One way you can claim superiority over Chu et al is by showing that in practical situations you can provide a more useful result. Hence, a comparison would help. Since you claim that your work is more general, perhaps you can test ExplaiNN on the kind of networks Chu et al consider (affine layers with relu activation) and show how your rules compare to their hyperplanes?

---

> > > ### Author Response · Authors · 2020-11-19
> > > **Response to Reviewer 2**
> > >
> > > Maybe we did not make this sufficiently clear in our previous response, but these two approaches are *not* applicable to state of the art networks, and hence a meaningful comparison is simply not possible. To re-iterate, method [1] and related approaches operate on Deep Belief Networks *only*, while method [2] is limited to very small (5-20 neurons per layer, very few hidden layers overall) piecewise linear networks *only*; neither are applicable to networks deployed in any real world setting. We simply *cannot* run these methods on e.g. VGG-S, GoogLeNet, ResNet, or other state-of-the-art networks.
> > >
> > > There are other key differences that render it hard to meaningfully compare to [1] *even* if it would run on state-of-the-art networks:
> > > (1) it is limited to single neuron antecedents, whereas we allow for multiple units, as well as multiple units that act in disjunction. Almost all results we show make use of these more complex rules, as they allow to discover shared and distinct trends across and within classes, but also to build these traces, which require multiple hidden units to be expressed by the antecedents.
> > > (2) The rule language is limited to conjunctions, which is not sufficient for noisy real world networks, or when considering output labels which are inherently mutually exclusive.
> > > (3) Most importantly, as stated before, the rules are based on weights *only*, hence they do not consider patterns of activations, which means they completely miss on the locality and how neurons act in concert.
> > > Next, even if we accept these differences, and assume we can run [1] on a state-of-the-art network, the question is how to compare the results. While we understand that it's always easier to determine which solution is 'best' by comparing numbers, quantifiying differences should only be done when *meaningful*.
> > > Simple counting of rules is no measure of quality, while comparing the found rules themselves does not make sense due to the differences in what they (can) express. Comparing confidence is also flawed, as ExplaiNN automatically discovers rules that expose high confidence in the observed activation patterns, wheres [1] defines its own metric of confidence as an approximation of the incoming weights, which is entirely different than what has been established in the pattern mining community in the last 25 years. In short, even if we could somehow run [1], there does not exist a sensible metric to compare the approaches.
> > >
> > > Method [2] is even harder to meaningfully compare to, as its contribution is mostly theoretic in nature; while convex polytopes indeed allow guarantees on separating hyperplanes that characterize the input-output mapping of the entire network, these do not give insight in how information *flows through* the network, do not reveal what *concepts* (groups of) nodes encode, are not they easily *interpretable* by a domain expert, nor does the approach scale beyond very small networks -- that is, even if we ignore all these intrinsic differences, we can still not run it on any reasonably sized real-world network and would be stuck with evaluation on toy settings.
> > >
> > > Regarding the correctness of ExplaiNN, because we base it on the MDL principle, which has strong information theoretic foundations, we have an intrinsic connection to probability theory. There is a direct correspondence between the maximum likelihood distribution of the data and the model that yields the minimal MDL score [*]. In short, the MDL optimal model corresponds to the distribution most likely to have generated the data. As most MDL scores, ours is hard to optimize: it is neither sub-modular, monotone, nor does it fall in any other class of combinatiorial optimization for which we can give theoretic guarantees on the optimality of greedy search. In that sense we cannot give real-world guarantees. What we do know, however, is that any pattern accepted into the model contributes to the overall compression (i.e. translated to hypothesis testing, it is significant with regard to the alpha set by the model cost). As such, any pattern that ExplaiNN returns is 'correct', ie. neither spurious, nor redundant. Our results show that we find useful representations of the data that are directly interpretable and yield novel insights. We provided a study with synthetic data with known ground truth in the revised manuscript.
> > >
> > > [*] J. Rissanen. Modeling by shortest data description.Automatica, 14(5):465–471, 1978

---

### Official Review · AnonReviewer4 · 2020-10-28
**The proposed method for black-box model exploration/inspection is sensible and provides a good evaluation, but I am not yet convinced of the contributions, distinction to related works and the empirical evaluation.**

**Rating:** 6
**Confidence:** 4

**Review:**

The paper proposes an approach to explainable supervised learning by extracting sets of rules for two individual layers within a neural network. The authors build their work on recent published work for patttern-based rule mining [0] to efficently find so-called robust rules. The authors evaluate the approach for image processing tasks with convolutional neural networks on MNIST, ImageNet and Oxford Flower by comparing generated rules against activation maps and prototypes.

The proposed approach is interesting, but I am left with several concerns and open questions with respect to contributions, available related works and the conducted evaluation.

While the paper discusses quite nicely summarized several relevant related works in the introduction section, I am still wondering how the approach is related to OpenAI's Circuits [1], where already individual neurons and connections between neurons are studied with respect to interpretability/explainability of neural nets? This is not to say that prior work already learned sets of rules among layers, but - if relevant - it should be evaluated to what extent the proposed method is superior for explaining neural nets to end-users. A smaller comment to the related work is that there might be missing references for model destillation, such as [2].

The taken approach to rule generation follows recent work on association rule mining, which is sensible. To this end, the paper misses to clearly address the difference to GRAB, i.e. the referenced algorithm used for learning the rules using model description length (MDL). Could you therefore elaborate why GRAB cannot be applied to the rule mining task out-of-the-box or clearer state if this what you have actually done? I feel like the presentation of the paper could benefit from answering this question, one could  establish a background section and/or focus on novel aspects of the proposed algorithm. In addition, there might be more space for the evaluation, which should be a major part of the paper's contribution, especially explainable ML.

The idea of exploring neural connections among two layers is generally intriguing. The paper could benefit from better motivating this design choice, i.e. why use "only" two layers? Would it make sense to use more than two convolutional layers for deeper networks? Would this be computationally tractable? Are two layers better to interpret for humans?

While the evaluation is insightful, I am not convinced that single-neuron-prototypes and activation maps are representative for all ongoing works on explainable and interpretable ML. More specifically with respect to activation maps, is it possible to rule out all activation map approaches for CNNs at once? What about perturbation-based approaches (e.g. [3])? To this end, I am wondering why you only compare to single neuron prototypes and not more complex prototypes of the individual class (for Sec. 3.1 and the MNIST experiment)? Lastly, I am also wondering if a user study would be helpful to confirm that the proposed explanations provide added value for end-users.

A minor comment is that it is not helpful for the reader that numerous references figures are in the appendix (e.g. the comparison of generated MNIST rules in Sec. 3.1 to the prototype in the appendix). A minor question would be why you define "prototype" as on a single-neuron-basis (in the introduction)? Is there a reference for the approach?

References:
[0] Fischer, J. and Vreeken, J., 2019, September. Sets of robust rules, and how to find them. In Joint European Conference on Machine Learning and Knowledge Discovery in Databases (pp. 38-54). Springer, Cham.
[1] Olah, C., Cammarata, N., Schubert, L., Goh, G., Petrov, M. and Carter, S., 2020. Zoom In: An Introduction to Circuits. Distill, 5(3), pp.e00024-001.
[2] Bastani, O., Kim, C. and Bastani, H., 2017. Interpreting blackbox models via model extraction. arXiv preprint arXiv:1705.08504.
[3] Fong, R.C. and Vedaldi, A., 2017. Interpretable explanations of black boxes by meaningful perturbation. In Proceedings of the IEEE International Conference on Computer Vision (pp. 3429-3437).

### Update after author response ###
Thanks again for the clarifications - After reading the author responses, the other reviewers' comments and the new version of the manuscript, I increase my score for the paper, as the authors now better state the relationship to Circuits and GRAB, and provide a significantly improved evaluation. The enhanced experimental section is now adequate for the paper's claims and offers additional insights into the usability of the generated rules.

---

> ### Author Response · Authors · 2020-11-13
> **Response to concerns of Reviewer 4**
>
> Thank you for this extensive review. In the following, we will adress all your concerns paragraph wise.
>
> The work by Olah et al. is actually one of the main motivations for our work. Their approach is best summarized in their own words: "What if we were willing to spend thousands of hours tracing through every neuron and its connections?", as they propose to painstakingly analyze prototypes of individual neurons, sort them, and see how information flows between them. With ExplaiNN we offer a method to instead do this automatically: in our GoogLeNet experiments we use their prototyping on the results of ExplaiNN -- without needing to spend thousand of hours to find the right connections.
>
> Furthermore, in their work they limit themselves to prototypes of single neurons. Groups of neurons that together encode useful  information are not considered at all. Clearly, as the task for individual neurons already takes "thousands of hours", the combinatorial explosion that occurs when we consider multiple neurons makes investigation by hand impossible. With ExplaiNN, however, we can automate this search, discover complex interactions between sets of neurons, and combine these with their (and those proposed by others) prototyping mechanisms -- which is what we did in our experiments.
>
> And thank you for the reference, it indeed slipped our attention. We add it to the upcoming revision.
>
> The second raised concern is how different ExplaiNN is to GRAB.
> First, why is GRAB insufficient here? There are multiple issues, which lead to the same consequence. GRAB's pattern language, considering only conjunctions, is too restrictive. Output neuron activations, for example, are often mutually exclusive; and hence GRAB would not be able to discover trends shared between classes. Hidden neuron activations, for example, are often noisy (see App. Fig. 7); and hence GRAB would only be able to model the noise-free parts of the neural dependencies. To alleviate this problem, we had to extend the GRAB pattern language to allow for disjunctions, which in turn required new optimization algorithms. Moreover, while GRAB scales well in n (samples) it does not scale well enough in m (neurons) and hence we had to additionally develop a new candidate generation scheme. We will clarify these aspects in the main body of the upcoming revision -- some of these were already included in the appendix for reasons of space.
>
> Regarding the third concern, on "restricting" to two layers, we first note that both method and theory are defined for two arbitrary sets of neurons, without specifying what or where these neurons are. In principle, we can mine rules between multiple layers without any restriction. While this might be of interest to some, we are particularly interested in how information flows through the network, and hence considered rules between consecutive layers, as this is how NNs pass on information. For ResNet-like architectures, however, we agree it would be interesting to examine not only the previous, but the previous two layers. Currently, ExplaiNN scales well enough to discover rules over many thousands of neurons within only a few hours. While this is generally sufficient to consider multiple layers at once, considering a full network at once might be a stretch: speeding up ExplaiNN further, which will likely include engineering tricks as well as clever new heuristics will make for interesting future research.
>
> On the fourth concern, we agree that the work of Fong et al is interesting, but note that it neither explains how neurons within the network interact nor how they are related to classes; they focus on saliency maps, where the goal is to retrieve those parts of a specific given input image that drives the classification.
> Activation maps on the other hand are applicable to hidden neurons, rather than input images, and are used in analysis of neural networks. We compare only to single neuron prototypes because this is the state of the art (see also Olah et al), and figuring out which neurons actually interact is exactly the task that ExplaiNN solves.
>
> We strongly agree with the final comment and will adapt the revision accordingly – we have many more interesting results than fit within the page limit. We plan to use the additional page to resolve this in upcoming revision.

---

> > ### Comment · AnonReviewer4 · 2020-11-20
> > **Response**
> >
> > Thank you for the clarifications. The differences to the original GRAB algorithm and, more importantly, the need for extending GRAB make sense are now elaborated on in the paper. You also included references to Circuits.
> >
> > After reading the other reviews, the concern with respect to the evaluation seems to remain. It was mentioned that it is rather qualitative and does make it difficult to generalize due to a small sample size. I can support this perception and also mentioned that a user study would be beneficial when independently analyzing the resulting set of rules / prototypes.
> >
> > While a user study might not the best fit for the venue, there might be other possibilities to evaluate the rules in specific contexts/applications of interest! To this end, there is very recent work on explaining neuron behavior with logical rules [1], which evaluates according to such dimensions. Would it be valuable to evaluate against such an approach? An interesting dimension for the evaluation then might be how well the investigated neurons actually support correct predictions, linking interpretabilty with performance. This might be an indirect, quantitative opportunity to validate the rules.
> >
> > Finally, can you give some statistics on how many rules are generated by the system for the particular tasks? Sorry if I overlooked this information.
> >
> > [1] Mu, J. and Andreas, J., 2020. Compositional explanations of neurons. Advances in Neural Information Processing Systems, 33.

---

> > > ### Author Response · Authors · 2020-11-24
> > > **Response**
> > >
> > > Thank you for this active discussion, we appreciate that you spend the time to give the constructive critique. Especially your proposal for a quantitative evaluation definitely improved the manuscript - thank you very much for that.
> > >
> > > In response to reviewer #1 we ran synthetic data experiments, to quantitatively evaluate how well ExplaiNN is able to retrieve rules from a data set with known ground truth, which can be found in the revised manuscript. With precision and recall regarding matching the ground truth set, we can assess the performance quantitatively. Your proposal for using prediction performance as a proxy to assess how good the rules are, and the great link to the paper you gave, inspired for an experiment that allows to quantify goodness of rules for a real world network, by doing an intervention on the weights connecting the head and tail of a rule. Great suggestion! After intervention we observe a tremendous drop in performance, often to 0 for the particular class. We further confirm that the rules are not spurious by running a randomized version of the intervention by setting an equal amount of weights towards that class to 0 at random. This makes a strong argument that the rule set as a whole is of high quality and corresponds to class specific knowledge that could even be used adversarially.
> > >
> > > The very recent proposal of Mu et al, is interesting, however very different in nature. It assumes that concepts are *known* and given as input to the algorithm, for which it then tries to optimize for neurons that express a co-activation in the input labeled with the particular concept, whereas ExplaiNN is entirely unsupervised and automatically discovers rules/concepts. This is also a key difference in terms of what we aim to find, as we explore what concepts a network learns unbiasedly, rather than if the learned concepts match a certain assumption or human intuition about a conceptual property of the input.
> > >
> > >
> > > With respect to the numbers, due to space problems we just mentioned that there are "several hundred up to few thousands of rules", to be more precise, for VGG-S on imagenet we got 3355, 811, respectively 1591 rules from output to fc7, fc7 to fc6, and fc6 to conv5 layer, for VGG-S on flowers we got 2324 rules for output to fc7, for MNIST we got 1447 for output to conv1, and 1592 for output to conv2. It is also easy to order these rules by compression gain to give the user guidance to the rules most important to explain the activations, if it is desired to only evaluate a subset of those.

---

### Official Review · AnonReviewer1 · 2020-10-28
**Review for Paper 3168**

**Rating:** 5
**Confidence:** 3

**Review:**

Paper Summary

The authors propose a method to explore how neurons interact within a neural network and derive rules of interactions that can help interpret the inner workings of the neural network and open up the black box. The algorithm, EXPLAINN, identifies rules between successive layers where each rule represents a set of neurons that are activate simultaneously and conditionally based on the previous layer. Minimum Description Length principle is used to derive an objective that minimizes the number of bits used to encode the rules. The rule sets are identified using a greedy heuristic and improved until convergence of the objective. The algorithm is then evaluated to demonstrate the interpretation of images with MNIST, GoogLeNet and VGG-S.


Positives
* The problem at hand is clearly important and can help with interpretation of the neural networks.  This is particularly important in fields such as biology (genomics) where better interpretation is often times desired at a slight cost in performance
* The formulation using Minimum Description Length Principle is definitely an interesting idea - particularly formulating the objective as set of independent and robust rules
* The framework is flexible and allows for discovery of rules relevant for subsets / combinations of classes and not just individual examples or global rules across examples
* The paper in general is well written and I particularly liked how the authors guide the readers through the different aspects of the algorithm evaluation. I particularly liked Appendix A since the concrete example made the objective function clear. The notations however can be better laid out and clarified.


Concerns
* My primary concern is the use of greedy heuristic to identify the rule set. This necessitates that each interaction individually carry some degree of information for a robust composite rule. While the rules identified through the heuristic could be informative, it is not clear whether they are the best set of rules since there are no bounds or guarantees of how the heuristic relates to the global optimum.
* On a related note, I felt the evaluation presented by authors while extensive is rather qualitative in nature. While the prototypes of identified rules across different datasets look relevant and interesting, it is not clear whether they are the best set of rules. So, I believe this manuscript needs benchmarking in datasets constructed with known rules (possibly through simulation) to alleviate these two major concerns
* The quantitative outputs of the ReLU activations are binarized for rule learning - the authors have not addressed how this impacts the accuracy of the rules since that quantitative information is used by the subsequent layers
* The impact of the threshold parameters theta and mu (Algorithms 1 & 2) are also not addressed - this seems critical to me since these thresholds define the initial set of rule candidates.

---

> ### Author Response · Authors · 2020-11-13
> **Response to Reviewer 1**
>
> Thank you for the constructive critique, this certainly helps to improve the manuscript.
>
> Regarding your first two concerns:
> We agree that quality guarantees or even optimal search are desirable, however, as our MDL objective is neither monotone nor sub-modular, and as the search space is exponential, the only available option is to employ heuristics. Our bottom-up strategy is based on the idea that a good large pattern, i.e. one that improves compression when we add it to our model, is composed of smaller parts that all help to compress the data. This is a commonly employed, succesful approach in pattern set mining, see e.g. GRAB [1], SLIM [2], SQS [3], kGIST [4]. One of the intuitions we exploit is that a larger rule will only lead to better compression if the `tails' of its generating rules co-occur under the same conditions, i.e. when the head of the rule is present. As if that is not the case, the new larger rule would leave many rows unexplained, and would hence likely not yield any gain.
> Before starting with the NN experiments, we validated this approach via initial experiments on synthetic data. We will update the manuscript with an extended version of these experiments as soon as they are finished.
>
> Regarding the third concern:
> It is correct that binarizing ReLU activations may result in information loss – activation strength might, for example, be used by subsequent layers – but by doing so we do gain a huge advantage in terms of interpretability: binarizing neuron activations allows us to derive crisp symbolic, and directly interpretable statements on how neurons interact. Moreover, the on/off state not only reflects how biological neurons function, but also closely matches tanh and sigmoid activation functions. Extending ExplaiNN to an easily interpretable pattern language that allows for continuous, ReLU-like, activations is an interesting line for future work.
>
> Regarding the fourth concern:
> Both these easily interpretable parameters are merely simple, yet effective runtime optimizations. From an MDL-perspective, the best results will always be obtained with the largest search space: i.e. with $\theta$ and $\mu$ set to 0, respectively $|X_1| + |X_2|$, as this corresponds to considering all combinations of rules. Besides impacting run-time, many of those rules may be uninteresting from a user-perspective. Mu and theta allow users to directly instruct ExplaiNN to ignore such rules.
> Here, we consider a good rule as one for which the tail items frequently occur when the head items are present. In other words, for a given rule $X \rightarrow Y$, if neurons $X$ are active, the neurons $Y$ should also likely to be active. This is reflected by the confidence parameter $\theta$, which is the chance that $Y$ is active when $X$ is active. In our experiments we found that rules of theta below 0.4 are too weak to reveal valuable insights, but this may be different for different users or domains.
> For the parameter $\mu$, we want to account for early rule merging decisions that hinder us to see a more general trend. For example it might be that we learned $ABC \rightarrow Y$  and $BCD \rightarrow Z$, which both individually make sense and give gain, yet the rule $BC \rightarrow YZ$ might be more general and yield even higher gain, but was not possible to obtain when the individual rules were considered. We used $\mu=6$, which allows to merge fairly large patterns with huge differences.
>
> [1] Fischer, J., Vreeken, J. Sets of Robust Rules, and How to Find Them. Proceedings of the European Conference on Machine Learning and Principles and Practice of Knowledge Discovery in Data (ECMLPKDD), Springer, 2019.
> [2] Smets, K., Vreeken, J. SLIM: Directly mining descriptive patterns. Proceedings of the SIAM International Conference on Data Mining (SDM), SIAM, 2012.
> [3] Tatti, N., Vreeken, J. The Long and the Short of It: Summarising Event Sequences with Serial Episodes. Proceedings of the ACM SIGKDD International Conference on Knowledge Discovery and Data Mining (KDD), ACM, 2012
> [4] Belth, C., Zheng, X., Vreeken, J., Koutra, D. What is Normal, What is Strange, and What is Missing in a Knowledge Graph. Proceedings of the Web Conference (WWW), ACM, 2020.

---

### Decision · Program_Chairs · 2021-01-07
**Final Decision**

**Decision:**

Reject

**Comment:**

This paper proposes a method to explore neuron interactions within a neural network by deriving rules for the activations of units at different layers. The rules can presumably help interpret the inner workings of the neural network.
The reviewers have very different opinions on the paper and the views did not converge.  However, there is a common concern on the lack of quantitative evaluation on the faithfulness of the rules to the models. I therefore do not recommend accept.

R1[5]: On a related note, I felt the evaluation presented by authors while extensive is rather qualitative in nature.
R2[3]: Given that I could provide you with a couple of references that you admit is relevant, and this was just off the top of my head, would you care to comment on a quantitative comparison with the referenced approaches?
R3[8]: The examples look very impressive, but my main concern is with whether the examples could have been cherry-picked, in the sense that most of the thousands of rules produced may not be useful.